# The conserved centrosomin motif, γTuNA, forms a dimer that directly activates microtubule nucleation by the γ-tubulin ring complex (γTuRC)

**Michael J Rale, Brianna Romer, Brian P Mahon, Sophie M Travis, Sabine Petry\***

Department of Molecular Biology, Princeton University, Princeton, United States

**Abstract** To establish the microtubule cytoskeleton, the cell must tightly regulate when and where microtubules are nucleated. This regulation involves controlling the initial nucleation template, the γ-tubulin ring complex (γTuRC). Although γTuRC is present throughout the cytoplasm, its activity is restricted to specific sites including the centrosome and Golgi. The well-conserved γ-tubulin nucleation activator (γTuNA) domain has been reported to increase the number of micro-tubules (MTs) generated by γTuRCs. However, previously we and others observed that γTuNA had a minimal effect on the activity of antibody-purified *Xenopus* γTuRCs in vitro (Thawani et al., *eLife*, 2020; Liu et al., 2020). Here, we instead report, based on improved versions of γTuRC, γTuNA, and our TIRF assay, the first real-time observation that γTuNA directly increases γTuRC activity in vitro, which is thus a *bona fide* γTuRC activator. We further validate this effect in *Xenopus* egg extract. Via mutation analysis, we find that γTuNA is an obligate dimer. Moreover, efficient dimerization as well as γTuNA's L70, F75, and L77 residues are required for binding to and activation of γTuRC. Finally, we find that γTuNA's activating effect opposes inhibitory regulation by stathmin. In sum, our improved assays prove that direct γTuNA binding strongly activates γTuRCs, explaining previously observed effects of γTuNA expression in cells and illuminating how γTuRC-mediated microtubule nucleation is regulated.

**\*For correspondence:**
spetry@Princeton.EDU

**Competing interest:** The authors declare that no competing interests exist.

## Editor's evaluation

This fundamental Research Advance is of interest to cell biologists studying the mechanisms and control of microtubule nucleation. Rale et al. convincingly establish the regulatory role of the γ-TuNA motif in microtubule nucleation and settle prior conflicting results in the literature. They show that γ-TuNA binds to and activates γ-TuRC-based microtubule nucleation both in *Xenopus* extracts and in vitro.

## Introduction

Microtubule (MT) assembly is a critical cellular process tightly regulated in both space and time. Spatio-temporal control of MT nucleation allows cells to use the same pool of soluble tubulin to generate different intracellular structures, from the interphase cytoskeletal transport network to the complex mitotic spindle. Yet, while the core MT nucleation machinery has been well characterized, how MT nucleation is locally activated remains poorly understood.

The key MT nucleator is the γ-tubulin ring complex (γTuRC). γTuRC is a large, 2.2 MDa complex that forms an asymmetric ring of γ-tubulin subunits (*Zheng et al., 1995*; *Moritz et al., 1998*). This ring is thought to act as an initial template for the MT (*Moritz et al., 2000*). As α/β-tubulin subunits

bind to the ring of γ-tubulin, they form the nucleus of a new MT, rapidly transitioning from nucleation toward the more favorable regime of MT polymerization (*Jackson and Berkowitz, 1980*; *Mitchison and Kirschner, 1984*). In vitro studies with purified human and *Xenopus* γTuRCs have shown that these can indeed catalyze the nucleation of new MTs (*Choi et al., 2010*; *Thawani et al., 2020*; *Liu et al., 2020*). Recent studies have also shown that γTuRC acts with the MT polymerase, XMAP215/ch-TOG, to nucleate MTs (*Thawani et al., 2018*; *Flor-Parra et al., 2018*; *Gunzelmann et al., 2018*; *King et al., 2020a*).

Structural studies of γTuRCs from yeast, frogs (*Xenopus laevis*), and humans revealed remarkable conservation of the γ-tubulin ring structure, although the composition of γTuRC differs substantially across these organisms (*Kollman et al., 2015*; *Liu et al., 2020*; *Wieczorek et al., 2020a*; *Wieczorek et al., 2020b*; *Consolati et al., 2020*). Intriguingly, the pitch and diameter of the γ-tubulin ring appears to be incompatible with that of the assembled MT lattice. This suggests that γTuRC undergoes a conformational change to reduce its diameter before it can nucleate MTs (*Thawani et al., 2020*; *Liu et al., 2020*). One possibility is that this activating conformational change is stimulated by direct binding of 'activation' factors. At the same time, other modes of activation are also plausible.

The centrosomal scaffold protein Cdk5rap2, which recruits γTuRC to the centrosome and Golgi (*Andersen et al., 2003*; *Bond et al., 2005*; *Fong et al., 2008*; *Choi et al., 2010*; *Mennella et al., 2012*; *Lawo et al., 2012*), has been shown to increase γTuRC's nucleation activity (*Fong et al., 2008*; *Choi et al., 2010*; *Roubin et al., 2013*). Previous domain-mapping studies found that the γ-tubulin nucleation activator (γTuNA or CM1) sequence in Cdk5rap2's N-terminus is critical to bind and activate γTuRC (*Figure 1A*; *Fong et al., 2008*; *Choi et al., 2010*). The γTuNA sequence is well-conserved across yeast, nematodes, flies, frogs, and humans (*Samejima et al., 2010*; *Conduit et al., 2014*; *Feng et al., 2017*; *Fong et al., 2008*; *Choi et al., 2010*), and identical γTuNA domains have been identified in related centrosomal and Golgi proteins such as myomegalin (*Roubin et al., 2013*). A bipartite version of γTuNA is also present in the microtubule branching factor, TPX2 (*Alfaro-Aco et al., 2017*; *King and Petry, 2020b*). Thus, understanding how the γTuNA domain interacts with γTuRC might bring insights into the regulation of MT assembly in a wide variety of organisms and contexts.

Direct binding of γTuNA has been proposed to activate γTuRC, as addition of γTuNA increases γTuRC activity in human cells (*Choi et al., 2010*; *Cota et al., 2017*). This activation effect in human cells is, in fact, also well-conserved across the phylogenetic tree with ectopic γTuNA expression triggering increased MT nucleation in fission yeast (*Lynch et al., 2014*), *Drosophila* (*Tovey et al., 2021*), and mice (*Muroyama et al., 2016*). Prior work has also identified a key hydrophobic residue in γTuNA, F75, that is critical for γTuNA's activation effect, suggesting a direct interaction with γTuRC involving this central region (*Fong et al., 2008*; *Choi et al., 2010*). Whether this activation effect is due to a direct increase in γTuRC activity has been an open question, although in vitro results with purified γTuRC and γTuNA suggest that this is the case (*Choi et al., 2010*; *Muroyama et al., 2016*). While these fixed endpoint results are suggestive, the field has been lacking a real-time, high-resolution observation of a direct γTuNA-mediated increase in γTuRC activity.

Previously we reported that γTuNA had little effect on the activity of antibody-purified *Xenopus* γTuRC (*Thawani et al., 2020*). Our observation was seemingly corroborated by independent in vitro and structural data published that same year (*Liu et al., 2020*). However, after substantial improvements in our γTuRC purification protocol, we now report the first real-time observation that the γTuNA domain directly increases γTuRC's nucleation ability. Using mutation analysis, we find that the γTuNA domain binds γTuRC as a dimer, providing the first biochemical validation of a recent γTuRC structural model containing a parallel coiled-coil binding partner presumed to be γTuNA (*Wieczorek et al., 2020a*). Critically, we show that complete dimerization of the γTuNA domain is required for binding and activation of γTuRC in extract and in vitro. Finally, we reveal that γTuNA-mediated activation of γTuRC is sufficient to counteract indirect regulation by the tubulin-sequestering protein, stathmin. In sum, our study provides a direct observation of γTuNA domains as *bona fide* γTuRC activators.

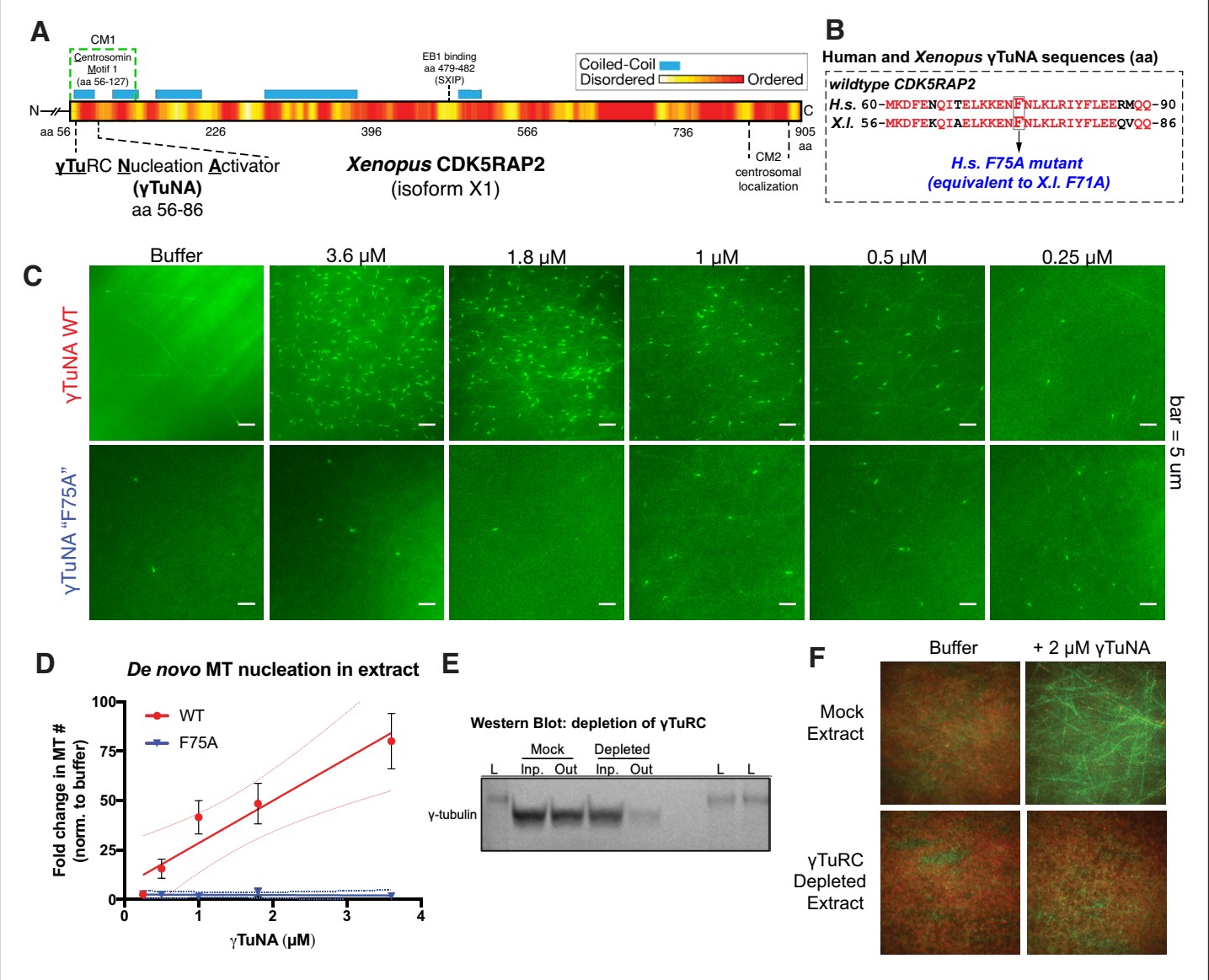

**Figure 1.** Cdk5rap2's γTuNA domain increases MT nucleation in *Xenopus* egg extract and requires the universal MT template, the γ-tubulin ring complex (γTuRC). (**A**) Schematic of *Xenopus* Cdk5rap2's domains. The γTuRC nucleation activator domain, γTuNA, is located from amino acids 56–86 in *Xenopus laevis* isoform X1 (905 aa) or 60–90 in human CDK5RAP2 isoform A (1893 aa). Predictions of disorder (PONDR-FIT; *Xue et al., 2010*) and coiled-coil regions (COILS) are shown as a red/yellow gradient or blue boxes, respectively. (**B**) Alignment of wildtype human and *Xenopus* γTuNAs. Identical residues are red. The human F75 residue (first mutated in *Fong et al., 2008*) is equivalent to residue F71 in *Xenopus*. In this study, mutations of well-conserved, identical residues are designated according to the human residue number (e.g. human F75A is equivalent to *Xenopus* F71A; both hereafter referred to as "F75A"). (**C**) TIRF assay of MT nucleation in *Xenopus* egg extract. A titration series of wildtype or 'F75A' versions of *Xenopus* γTuNA (Strep-His-Xen. γTuNA-aa 56–89) were added to extract as shown. EB1-mCherry was used to mark growing MT plus-ends (pseudo-colored green in images). Bar = 5 μm. (**D**) Quantification of the number of EB1 spots in C. The data were normalized by the buffer controls, and are shown as fold-changes. Black error bars are the standard error of the mean (SEM) for three independent extracts. Thin colored lines on either side of the central trendline represent 95% confidence intervals. (**E**) Western blot of γ-tubulin levels before and after mock-treatment or incubation with Strep-His-Halo-*Xenopus* γTuNA-coupled beads. After a single pulldown, the majority of γ-tubulin signal is lost. (**F**) TIRF assay of mock- and γTuRC-depleted extract. Alexa-488 labeled tubulin (green) and EB1-mCherry (red) were used to visualize microtubules in extract with or without 2 μM Strep-His-*Xenopus* γTuNA. See "*Figure 1—source data 1*" and "*Figure 1—source data 2*" for numerical data and raw blot.

The online version of this article includes the following source data and figure supplement(s) for figure 1:

**Source data 1.** Numerical data for *Figure 1*.

**Source data 2.** Labeled and raw blots used in *Figure 1*.

**Figure supplement 1.** Different sizes of *Xenopus* γTuNA bind γTuRC with different affinities.

*Figure 1 continued on next page*

*Figure 1 continued*

**Figure supplement 1—source data 1.** Labeled and raw blots used in *Figure 1—figure supplement 1*.

**Figure supplement 2.** Addition of γTuNA to *Xenopus* egg extract does not affect γTuRC assembly or stability.

**Figure supplement 2—source data 1.** Labeled and raw blots used in *Figure 1—figure supplement 2A and C*.

## Results

### Cdk5rap2's γTuNA domain increases MT nucleation in *Xenopus* egg extract

To study how *Xenopus* Cdk5rap2 affects γTuRC's activity, we added its purified γTuNA domain (*Figure 1A–B*; *Figure 1—figure supplement 1*; aa 56–89, isoform X1) to *Xenopus laevis* egg extract and assessed its impact on microtubule (MT) nucleation (*Figure 1C*). Using total internal reflection (TIRF) microscopy and fluorescent end binding protein 1 (EB1) to label growing MT plus ends, we quantified individual MT nucleation events (*Figure 1C–D*). In the control reaction, the egg extract showed a typical low level of MT nucleation (*Figure 1C*, 'buffer', ~3 MTs per field). In contrast, addition of wildtype γTuNA triggered an increase in MT nucleation of up to ~75-fold in a titration series (*Figure 1C–D*). The *Xenopus* F71A mutant equivalent to the human F75A mutant (*Figure 1B*), hereafter referred to as 'F75A', did not significantly increase MT number even at the highest concentration (3.6 µM, *Figure 1C–D*). Thus, the γTuNA domain activates MT nucleation in extract and requires the F75 residue, validating prior studies (*Fong et al., 2008*; *Choi et al., 2010*). Using sucrose gradients to fractionate mock and γTuNA-treated extracts, we also conclude that the γTuNA domain has no effect on γTuRC assembly, ruling out one possible explanation for this increase in MT number (*Figure 1—figure supplement 2*). While we cannot rule out that full-length Cdk5RAP2 might affect γTuRC assembly, we believe this is also unlikely as recent work has demonstrated that γTuRC can be assembled via heterologous expression of just γTuRC components and the RUVBL1-RUVBL2 AAA ATPase complex, without addition of a CM1-containing protein (*Zimmermann et al., 2020*).

### The γTuNA domain requires the universal MT template, the γ-tubulin ring complex (γTuRC)

We next confirmed whether the γTuNA domain's ability to increase MT nucleation in extract was dependent on the known MT nucleator, γTuRC. To do this, we first attempted depleting γTuRC from extract using our previously published rabbit-derived, anti-gamma tubulin antibody (*Thawani et al., 2020*). This γTuRC-depleted extract would then be assayed in the presence of γTuNA in our TIRF assay. However, due to low antibody yields and batch-to-batch variability, we were unable to generate γTuRC-depleted extract at consistent levels via this method. As an alternative, we instead depleted extracts of γTuRC via pulldown of γTuNA-coupled beads. With a single round of depletion, we observed a loss of >75% of γ-tubulin signal indicating a depletion of γTuRC (*Figure 1E*). In the mock-treated extract where γTuRC was not depleted, the γTuNA domain's ability to increase MT nucleation levels remained unchanged (*Figure 1F*). By contrast, exogenous γTuNA no longer activated MT nucleation in γTuRC-depleted extracts (*Figure 1F*). Hence, the γTuNA domain requires the universal MT template, γTuRC, to activate MT nucleation.

### The γTuNA domain can designate new artificial MTOCs by recruiting γTuRC

As γTuNA co-depletes γTuRC, we wondered whether this interaction would be sufficient to generate artificial MT asters (*Figure 2A*). To that end, we coated micron-scale beads with wildtype or mutant γTuNA domains and added them to extract. After a pulldown step, we assayed these beads for MT aster formation in vitro in the presence of purified fluorescent tubulin and GTP under oblique TIRF (*Figure 2B*). We found that wildtype γTuNA-coated beads formed large MT asters mimicking the potent MT nucleation of the centrosome (*Figure 2B*). In contrast, the F75A mutant beads formed severely impaired asters (*Figure 2B*). Mock-treated beads did not form asters. To confirm the stable presence of γTuRC, we repeated the bead pulldown from extract and attached the beads via an antibody against Mzt1, a γTuRC subunit, to surface-treated coverslips (*Figure 2A*). We then added fluorescent tubulin and GTP before live imaging via TIRF microscopy (*Figure 2C*). Critically, we observed

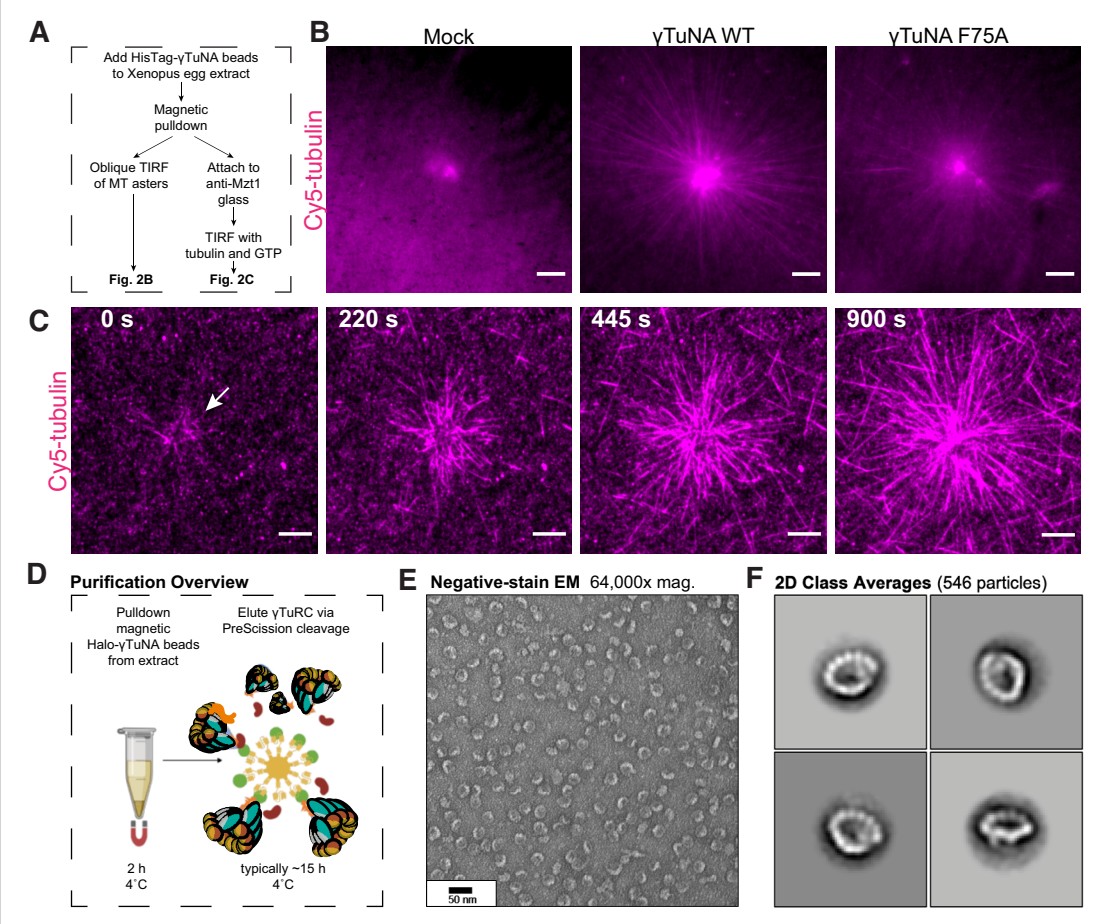

**Figure 2.** *The γTuNA domain strongly recruits MT nucleation factors (including γTuRC) from Xenopus egg extract.* (**A**) Schematic of experiments for B and C. (**B**) Oblique TIRF images of MT asters from beads in vitro after 10 min. HisPur magnetic beads coated with either bovine serum albumin (mock), Strep-His-*Xenopus* γTuNA wildtype (WT), or 'F75A' mutant were incubated with extract, pulled-down, and washed. These were then diluted 1/1000 with polymerization mix containing 15 μM tubulin and 1 mM GTP, before imaging with TIRF. 5% Cy5-tubulin was used to label MTs. Bar = 5 μm. (**C**) Time-lapse imaging of MT aster growth from wildtype γTuNA beads in vitro. As in part B, wildtype γTuNA beads were pulled-down from extract and washed. These were then incubated on DDS-surface treated coverslips coated in anti-Mzt1 antibody to attach beads containing γTuRC. After a wash step, polymerization mix was added prior to time-lapse TIRF imaging. Frames are shown over the course of 15 min (900 s). Bar = 5 μm. (**D**) Diagram showing purification of endogenous *Xenopus* γTuRC using magnetic beads coupled to Strep-His-HaloTag-3C-human γTuNA. Made partly with Biorender. (**E**) Representative image of purified γTuRCs via negative-stain electron microscopy. Magnification is 64,000 x, taken at 80 kV with a Philips CM100 transmission electron microscope. Bar = 50 nm. (**F**) 2D class averages of 546 γTuRC particles picked from negative-stain EM images like in E. Each image represents one of four top classes. See "*Figure 2—source data 1"* for uncropped images in B and E.

The online version of this article includes the following source data and figure supplement(s) for figure 2:

**Source data 1.** Uncropped images for *Figure 2*.

**Figure supplement 1.** Mass spectrometry (Quant-IP) reveals γTuRC is the dominant factor present after extract pulldown of Halo-γTuNA beads.

**Figure supplement 1—source data 1.** Labeled and raw blots used in *Figure 2—figure supplement 1B and C*.

**Figure supplement 1—source data 2.** Raw mass spectrometry data for pulldowns of Halo-γTuNA from *Xenopus* egg extract (TCMP- ProQuant).

**Figure supplement 2.** Purity and concentration assessment of γTuRCs purified via Halo-γTuNA pulldown.

**Figure supplement 2—source data 1.** Labeled and raw blots used in *Figure 2—figure supplement 2A, B, C*.

**Figure supplement 3.** γTuRCs purified via Halo-γTuNA pulldown are fully assembled rings.

that wildtype γTuNA beads attached and formed large MT asters in vitro, indicating that these beads had retained γTuRC and any other necessary MT nucleation factors (*Figure 2C*, *Video 1*).

From this we conclude that γTuNA domains are sufficient to specify new sites of γTuRC-mediated MT nucleation. Critically, this finding allowed us to develop a new γTuRC purification scheme based

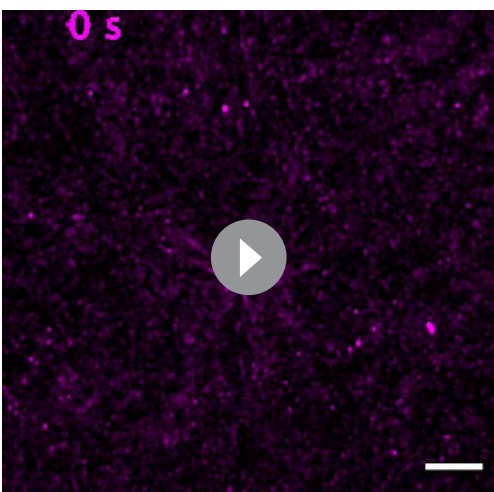

**Video 1.** Post-pulldown wildtype γTuNA beads nucleate asters in vitro.
https://elifesciences.org/articles/80053/figures#video1

on scaled-up pulldowns with Halo-human γTuNA (outlined in *Figure 2D–F*), which we discuss in more detail later. Finally, the ability of F75A beads to weakly nucleate asters points to residual, but persistent, binding of γTuRC. We believe this is due to the low stringency wash particular to this experiment, as we do not detect γTuRC on F75A beads after higher stringency washes in subsequent experiments (western blots in *Figure 2—figure supplement 1C*, *Figure 3D–E*).

## γTuNA is an obligate dimer

Having confirmed that the γTuNA domain strongly recruits γTuRC from extract, we next investigated the γTuNA-γTuRC interaction. In a recent structural study, the authors generated a model of a parallel coiled-coil that directly interacts with γTuRC (*Wieczorek et al., 2020a*). The authors suggested that this coiled-coil is in fact a γTuNA dimer, although biochemical validation of this dimer state and its effect on γTuRC activity were not provided (*Wieczorek et al., 2020a*). To that end, we selectively mutated hydrophobic residues found within a heptad-repeat region of γTuNA. Specifically, we mutated the hydrophobic residues F63, I67, L70, and L77 to either alanine or aspartate (*Figure 3A*). To validate the well-conserved nature of this domain, we generated both human and *Xenopus* versions, referred to here by the residue position in the human sequence (*Figure 1B*).

We initially focused on the double, triple, and quadruple mutants for both human and *Xenopus* γTuNAs. We performed size-exclusion chromatography (SEC) and compared the peak retention volumes of wildtype and mutated γTuNAs. Our SEC data revealed that wildtype γTuNA is a dimer (*Figure 3B–C*). By comparing the SEC traces for the double, triple, or quadruple mutants from both *Xenopus* and human γTuNAs, we found that γTuNA dimerization was dependent on residues I67, L70, and L77 (*Figure 3B*). The double hydrophilic mutants (I67D/L70D) from both human and *Xenopus* versions were entirely monomeric. This was also true for the human double-alanine mutant, I67A/L70A (*Figure 3B*).

To resolve each residue's individual contribution to γTuNA dimerization, we generated alanine point mutants for F63, I67, L70, and L77 in *Xenopus* γTuNA. We also tested the F75A mutant of *Xenopus* γTuNA, as we wanted to know whether its loss-of-function coincided with loss of dimerization. We compared the SEC traces for these point mutants and found that mutating residues F63 or F75 to alanine had no deleterious effect on γTuNA dimerization (*Figure 3B*). By contrast, individually mutating residues I67, L70, or L77 increasingly interfered with dimerization, resulting in intermediate populations between full dimer and full monomer (*Figure 3B–C*). Mutation of the L70 or L77 residues resulted in the most drastic impairment, further confirming that this central region is crucial for γTuNA dimerization.

## Both dimerization of γTuNA and its F75 residue are critical for binding γTuRC

With the insight that the γTuNA domain is an obligate dimer, we next asked whether dimerization was required to bind γTuRC. We performed pulldowns of N-terminally Halo-tagged γTuNA mutants from *Xenopus* egg extract. We determined the amount of γTuRC bound for each γTuNA construct by probing for the γTuRC components GCP5 and γ-tubulin (*Figure 3D–G*). We found that both human and *Xenopus* double aspartate mutants (I67D/L70D), as well as the human triple mutant (I67D/L70D/L77D) did not bind γTuRC, indicating that loss of dimerization results in loss of γTuRC binding (*Figure 3D and F*). Interestingly, we found that the intermediate dimer mutants (I67A, L70A, or L77A) had correspondingly intermediate levels of γTuRC binding ability (*Figure 3E*). The I67A mutant, for example, was only weakly impaired in terms of dimerization (*Figure 3B*) and subsequently retained

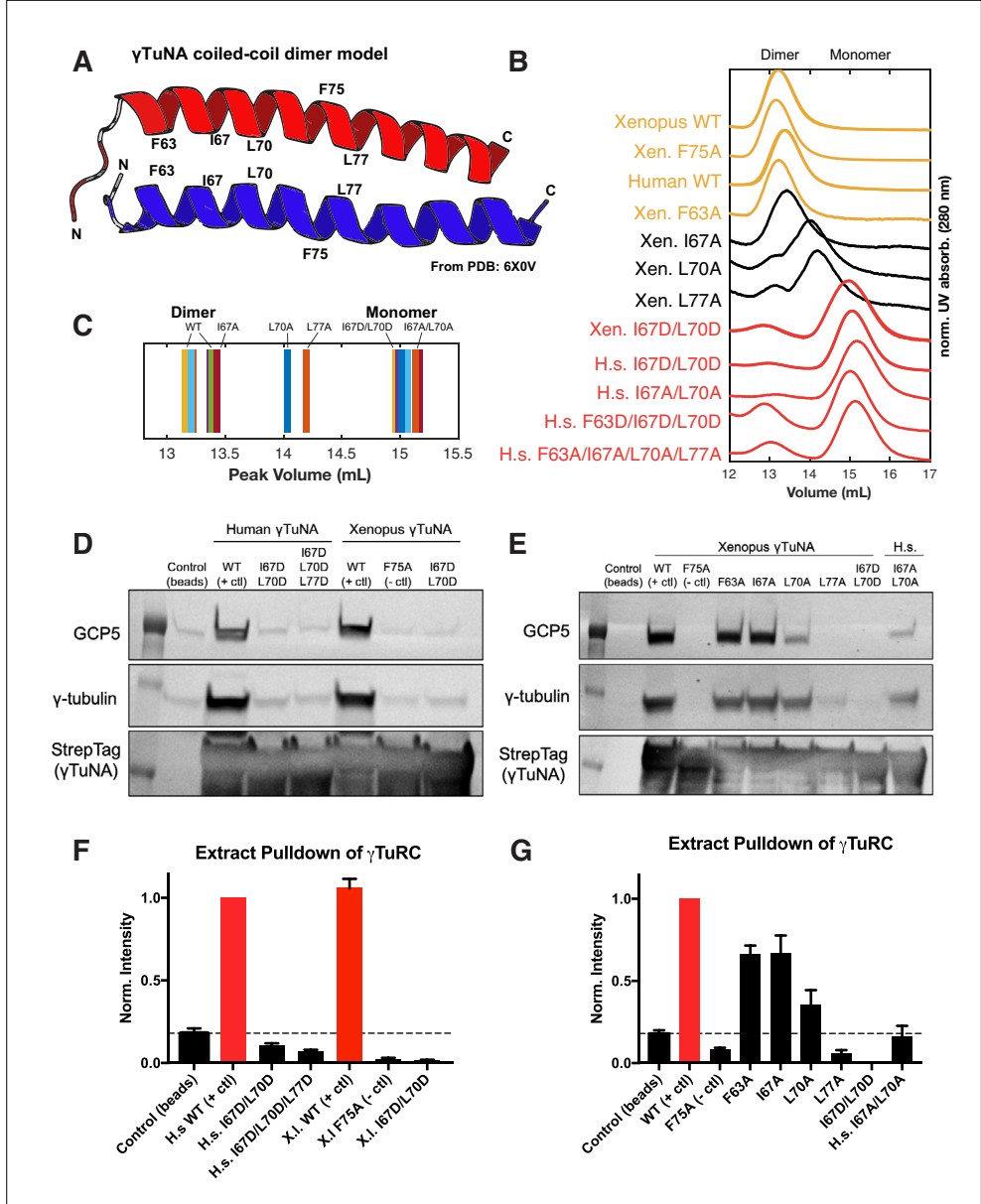

**Figure 3.** γTuNA requires both dimerization and the F75 residue to bind γTuRC in extract. (**A**) Model of dimerized, coiled-coil γTuNA with labeled side-chains for residues F63, I67, L70, F75, and L77. Made using PyMOL (RRID:SCR_000305) and chains C/G (red color) and D/H (blue color) from PDB: 6X0 V (*Wieczorek et al., 2020a*). (**B**) Size-exclusion chromatograms for human (aa 53–98) and *Xenopus* (aa 56–101) Halo-γTuNA wildtype and mutant constructs. Proteins were run at 50 μM (monomer) on a Superdex 200 increase 10/300 GL column (Cytiva) on an Äkta Pure system. Absorbance traces (A280 nm) were normalized by their peaks and plotted stacked as shown. (**C**) Diagram of peak retention volumes for each construct tested. (**D**) Western blots for γTuRC components, GCP5 and γ-tubulin, pulled down by beads coupled to human and *Xenopus* Halo-γTuNAs incubated in egg extract. The Strep-tag blot is shown as a bead loading control. (**E**) Western blots as in D, except comparing pulldowns done with Halo-*Xenopus* γTuNA alanine point mutants, with wildtype and F75A mutants as positive and negative controls. (**F**) Quantification of γTuRC pulldowns shown in D, normalized to the band intensity for human wildtype Halo-γTuNA beads. N=3. Error bars are SEM. (**G**) Same quantification of γTuRC pulldowns as in F, except for pulldowns as done in E. Normalized to the band intensity of wildtype *Xenopus* γTuNA. N=2. Error bars are SEM. See "*Figure 3—source data 1*" for numerical data and "*Figure 3—source data 2*" for raw blots.

The online version of this article includes the following source data and figure supplement(s) for figure 3:

**Source data 1.** Numerical data for *Figure 3*, includes normalized size-exclusion chromatography traces for *Figure 3B*, quantified pulldowns in *Figure 3F*, and quantified pulldowns in *Figure 3G*.

*Figure 3 continued*

**Source data 2.** Labeled and raw blots used in *Figure 3D and E*.

**Figure supplement 1.** *Rescuing γTuNA dimerization is not enough to rescue γTuRC binding.*

**Figure supplement 1—source data 1.** Labeled and raw blots used in *Figure 3—figure supplement 1B*.

its ability to bind γTuRC (*Figure 3E and G*). As dimerization was increasingly impaired in the L70A and L77A mutants, γTuRC binding became increasingly weaker (*Figure 3G*). In the most extreme example, the L77A mutant, which had the most substantial dimerization defect, had complete loss of γTuRC binding (*Figure 3G*). Critically, the known F75A mutant did not bind γTuRC, as expected (*Figure 3D–G*). As our SEC data shows that F75A does not affect γTuNA dimerization, we conclude that both γTuNA dimerization and the F75 residue are required for binding γTuRC (*Figure 3B and D–G*). Finally, we found that forcing γTuNA dimerization via the addition of a constitutively dimeric coiled-coil domain (GCN4) did not rescue the ability of the intermediate dimer mutants to bind γTuRC (*Figure 3—figure supplement 1*). This suggests that simply bringing intermediate dimer mutants within tight proximity is not enough to induce restoration of the proper γTuRC binding interface.

## Both γTuNA dimerization and the F75 residue are required for full γTuRC activation in extract

Having identified specific mutations that impaired γTuNA's ability to dimerize and bind γTuRC, we next asked what effect these mutants had on MT nucleation in extract. We added wildtype or mutant *Xenopus* γTuNA to freshly prepared extracts and again tracked MT plus-ends via fluorescent EB1 as a measure of MT number (*Figure 4*). As before, wildtype γTuNA triggered an increase in MT nucleation, when compared to the buffer control (*Figure 4*). The F75A mutant had little effect on extract MT levels (*Figure 4*). Similarly, the L77A mutant, which cannot bind γTuRC in

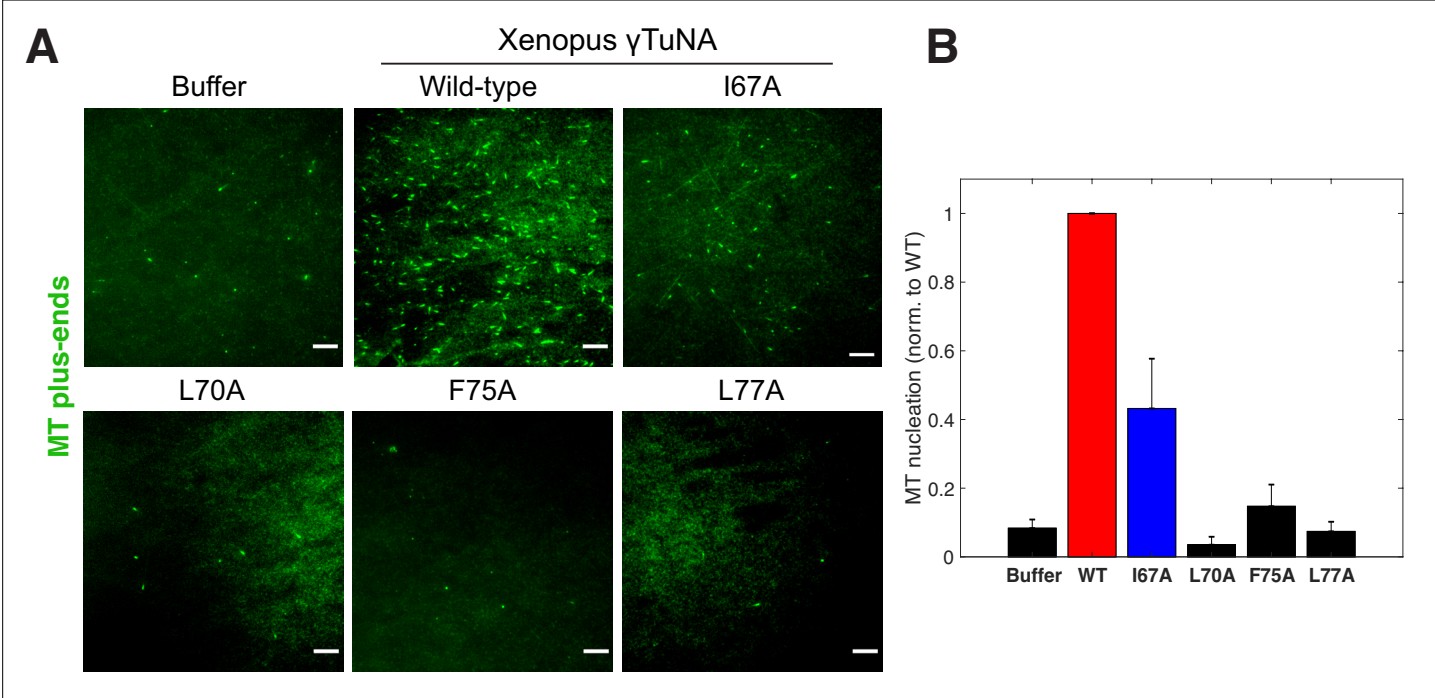

**Figure 4.** Complete γTuNA dimerization is required to maximally increase MT nucleation in extract. (**A**) TIRF assay of MT nucleation in extract after addition of 2 μM (1 μM dimer) wildtype or single alanine mutants of Strep-His-*Xenopus* γTuNA. EB1-mCherry was used to count MTs (MT nucleation) and is shown pseudo-colored green. Images were taken after 5 min at 18–20°C. Bar = 5 μm. (**B**) Quantification of MT nucleation (MT number) normalized by the wildtype condition across four independent experiments. Red bar denotes wildtype level, while the blue bar denotes the effect of the I67A mutant. Error bars are SEM. See "*Figure 4—source data 1*" for numerical data.

The online version of this article includes the following source data for figure 4:

**Source data 1.** Numerical data used in *Figure 4*.

extract (*Figure 3G*), did not increase MT nucleation (*Figure 4*). Intriguingly, when we examined the intermediate γTuRC-binding mutants I67A and L70A, we found that the I67A mutant activated MT nucleation to ~50% of wildtype levels, but L70A had no activity (*Figure 4B*). This was surprising as I67A's activation effect was on the order of its γTuRC-binding ability (~50% vs~67%; compared to wildtype), suggesting binding ability was predictive of the activation effect in extract (*Figure 3G*). However, because the L70A mutant had little activity in extract (~6%, *Figure 4B*) but retained ~35% binding ability (*Figure 3G*), it appears that there is a threshold to γTuRC's activation in extract. We further analyze the implications of this divergent behavior between γTuNA mutants in our Discussion.

## The γTuNA domain directly activates MT nucleation by γTuRC in vitro

While we had explored the effect of wildtype γTuNA and its dimer mutants on MT nucleation in extract, we had yet to determine if γTuNA directly increased γTuRC's activity in vitro. As we briefly mentioned (*Figure 2D–F*), we used beads coupled to a Halo-human γTuNA construct to purify endogenous *Xenopus* γTuRC from extract (*Figure 2D–F*), similar to previous work (*Wieczorek et al., 2020b*). Mass spectrometry confirmed that the dominant co-precipitant was indeed *Xenopus* γTuRC (*Figure 2—figure supplement 1*). We also confirmed the presence of fully assembled γTuRC rings via negative-stain electron microscopy (*Figure 2E–F* and *Figure 2—figure supplement 3*). Using this purified γTuRC, we investigated the effect of wildtype and mutant γTuNAs on γTuRC's activity in vitro via in vitro TIRF assays (*Figure 5*). In these assays, biotinylated γTuRCs were attached to passivated coverslips before imaging with TIRF microscopy (schematized in *Figure 5—figure supplement 1*). This not only offers high signal-to-noise but also allows tracking of individual γTuRC-mediated MT nucleation events.

We started by first comparing total MT mass generated in our assay (*Figure 5B*). Strikingly, the addition of γTuNA triggered a 5-fold increase in MT mass as compared to the buffer control (*Figure 5B*, *Video 2* and *Video 3*). To determine if this was a direct stimulation of γTuRC's activity, we then quantified the number of γTuRC-nucleated MTs within the first 150 s (*Figure 5C*), the MT nucleation rate (*Figure 5D*), the mean MT growth speed (*Figure 5E*), and the mean maximum MT length (*Figure 5F*). These quantifications revealed that wildtype γTuNA sharply increased the γTuRC nucleation rate from 1.2 MTs/s to 24.5 MTs/s (~20-fold increase, *Figure 5C–D*). While there was a slight increase in mean MT growth speed (+0.2 µm/min), this did not translate into a significant effect on MT length (*Figure 5E–F*). We also found that wildtype γTuNA saturated our assay within 30 s (*Figure 5C*), with a decreased nucleation rate of 0.15 MTs/s that remained constant for the remainder of the experiment (*Figure 5—figure supplement 1*). Thus, we conclude that γTuNA's effect is almost exclusively due to a direct ~20-fold increase in γTuRC activity and not due to altered MT dynamics.

Incidentally, we also found that bulky N-terminal tags on γTuNA completely ablated its ability to activate γTuRC, instead turning it into a specific γTuRC repressor (see *Figure 5—figure supplement 2*).

As expected, the F75A mutant did not increase MT mass in vitro (*Figure 5B*). Furthermore, both the F75A and L77A mutants had no significant effect on the initial γTuRC nucleation rate (*Figure 5D*). However, we did observe that L77A caused a weakly significant increase in MT number beginning at 150 s (*Figure 5C*, p ~ 0.03). Similarly, we found that around 150 s L77A increased γTuRC's nucleation rate 1.4-fold when compared to buffer (*Figure 5—figure supplement 1*). Beyond 150 s, the L77A mutant triggered a delayed increase in MT mass (*Figure 5B*), a behavior not observed in extract (*Figure 4*). From this discrepancy, we inferred that the presence of other factors in extract blocks impaired dimer mutants like L77A from interacting with or stimulating γTuRC.

To confirm that the γTuNA domain's activation effect could be modeled simply as a change in γTuRC activity, we simulated the above experiments using our experimentally determined nucleation rates. We found that a simple deterministic model based on the initial nucleation rates was sufficient to capture most of the behavior in our system, except for the L77A mutant of γTuNA (*Figure 5—figure supplement 3*). Using a single constant growth speed for all conditions, we found that the nucleation rate completely captured the effect of wildtype and F75A γTuNA on MT number and mass (*Figure 5—figure supplement 3*). For L77A, our simulation suggests a two-phase behavior where its effect on nucleation rate increases at some late stage, possibly due to a shift from an impaired dimer state to complete dimer when bound to γTuRC (*Figure 5—figure supplement 3F*). Regardless, both

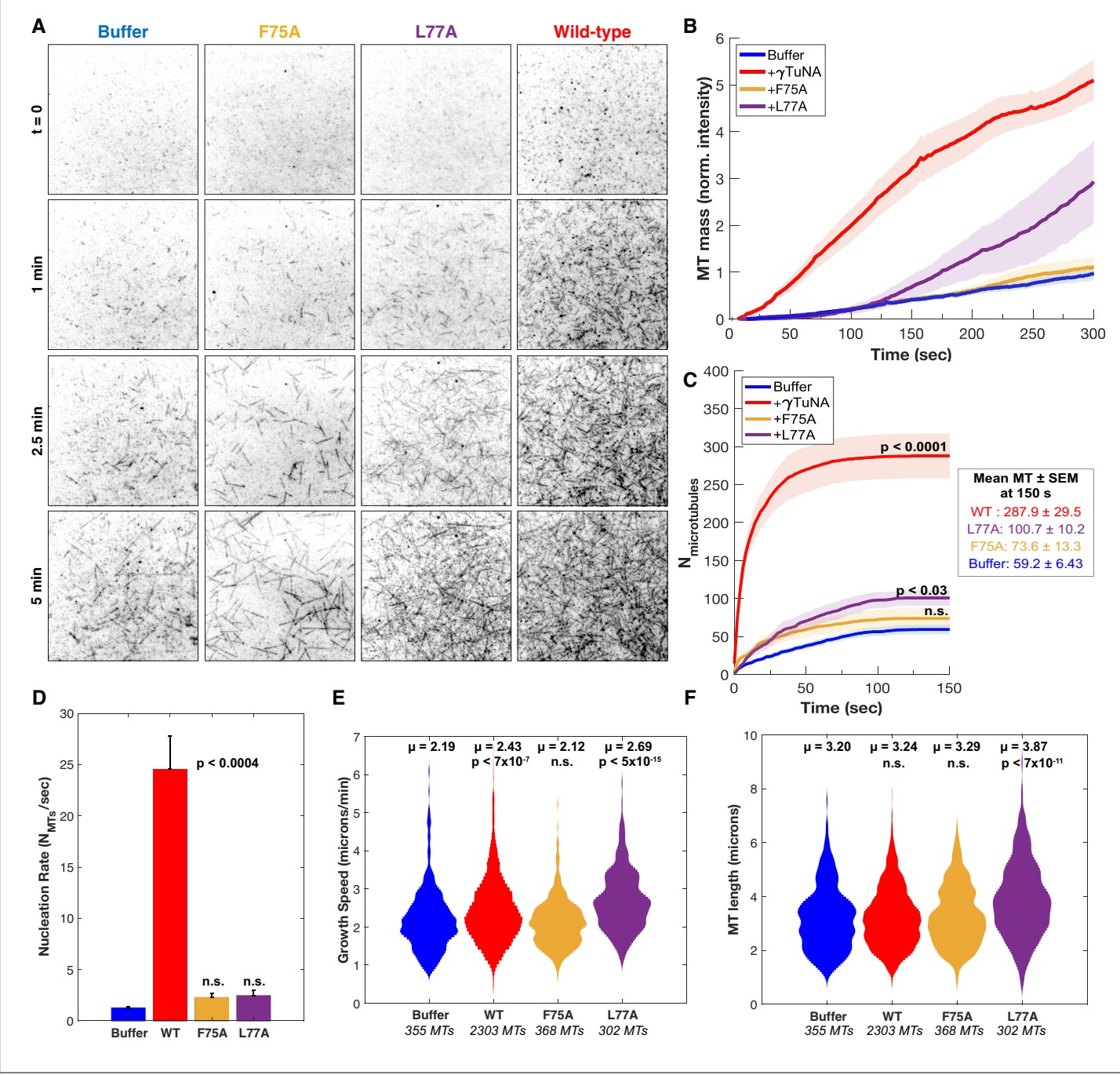

**Figure 5.** γTuNA dimers directly activate γTuRC MT nucleation ability in vitro. (**A**) Single molecule TIRF assays of γTuRC-mediated MT nucleation in vitro. Purified *Xenopus* γTuRCs were biotinylated and attached to passivated coverslips via surface-bound Neutravidin molecules. Polymerization mix containing 15 µM tubulin,1 mM GTP, and either control buffer or 3.3 µM (1.7 µM dimer) Strep-His-*Xenopus* γTuNA was then added. Wildtype, F75A, and L77A versions of γTuNA were tested. 5% Alexa 568-tubulin was used to visualize MTs. Images were taken every 2 s, for 5 min total, at 33.5 °C. Wildtype γTuNA (n=8), buffer control (n=6), γTuNA-F75A (n=5), and γTuNA-L77A (n=3). (**B**) Mean MT signal (MT mass) over time, normalized to the buffer condition at 300 s. (**C**) Mean MT number over time (measured for the first 150 s). The box shows the mean MT number ± SEM at 150 s for each condition. (**B and C**) Solid lines are the mean over time, with error clouds representing SEM. (**D**) Initial nucleation rates (Mts nucleated per sec) for each condition (± SEM). The curves shown in part C were fit to an exponential function to determine k (the nucleation rate). Each k was then averaged; see Materials and methods. The following are mean nucleation rate ± SEM. Buffer: 1.2±0.15 MTs/s, WT: 24.5±3.27 MTs/s, F75A: 2.3 ± 0.41 MTs/s, L77A: 2.4±0.55 MTs/s. (**E and F**) Violin plots of MT growth speeds (in **E**) or MT lengths (in **F**) for each condition. Means (µ) are shown alongside p-values. Wildtype γTuNA (n=2303 MTs), buffer control (n=355 MTs), γTuNA-F75A (n=368 MTs), and γTuNA-L77A (n=302 MTs). (**C-F**) Two-sample unpaired t-tests

*Figure 5 continued on next page*

*Figure 5 continued*

were used to compare the buffer control to the experimental values. Significance is p<0.05. See "*Figure 5—source data 1*" for all numerical data presented here.

The online version of this article includes the following source data and figure supplement(s) for figure 5:

**Source data 1.** Numerical data from *Figure 5*'s in vitro TIRF assays with purified γTuRC and γTuNA: including MT mass measurements, MT number, MT growth speed, and MT lengths.

**Figure supplement 1.** Overview and additional single molecule TIRF data.

**Figure supplement 1—source data 1.** Numerical data used in *Figure 5—figure supplement 1*; late-stage nucleation rates for experiments from *Figure 5*.

**Figure supplement 2.** The presence of large, bulky N-terminal tags on γTuNA directly inhibits γTuRC activity in extract and in vitro.

**Figure supplement 3.** Simulation of γTuNA's effect on γTuRC MT nucleation activity.

the in vitro and simulated data demonstrate that wildtype γTuNA's effect on MT nucleation is due to a direct increase in γTuRC activity without altering MT dynamics.

## γTuNA activation of γTuRC in extract overcomes the effect of negative regulators like stathmin

As there appeared to be an activation barrier in extract, but not in vitro, we investigated whether known negative regulators of MT nucleation were responsible. We focused on the tubulin-sequestering protein stathmin (or op18), which regulates the available tubulin pool for nucleation and polymerization ( *Belmont and Mitchison, 1996*; *Gavet et al., 1998*). For every mole of stathmin present (1.5 µM endogenous concentration), two moles of tubulin are removed (*Gigant et al., 2000*; *Wühr et al., 2014*).

We sought to first confirm that stathmin negatively regulated MT nucleation by γTuRC (as in our previous work; *Thawani et al., 2020*), and then determine whether γTuNA had any effect on this. To that end, we added increasing amounts of exogenous stathmin to extract and measured its effect on MT nucleation (*Figure 6A–B*). At double and triple the endogenous concentration of stathmin in extract (~3 µM or ~4.2 µM final), we observed a drastic loss of MT nucleation and polymerization (*Figure 6A–B*). Surprisingly, γTuNA was still able to activate MT nucleation (*Figure 6A*), even at the highest concentration of stathmin tested, with a ~ five-fold increase in the number of MTs (*Figure 6B*, 2.7 µM stathmin).

Next, we assessed whether the γTuNA-γTuRC complex could also overcome stathmin's effect in vitro (*Figure 6C*). For simplicity, we tracked the total MT signal (or MT mass) produced after 250 s in the TIRF-based nucleation assay (*Figure 6D*). We first observed the activity of γTuRC alone at 15 µM tubulin (*Figure 6C*, upper left panel). We next tested γTuRC in the presence of either 2.5 or 4 µM stathmin and found that stathmin resulted in losses of 59% and 81% MT mass, respectively (*Figure 6C*, 'No γTuNA' conditions). Interestingly, addition of γTuNA to stathmin and γTuRC rescued

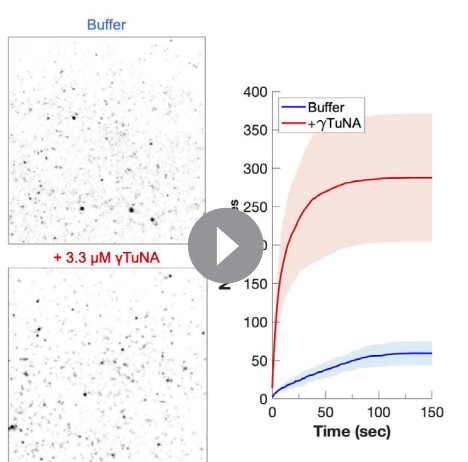

**Video 2.** Wildtype γTuNA directly stimulates γTuRC in vitro.

https://elifesciences.org/articles/80053/figures#video2

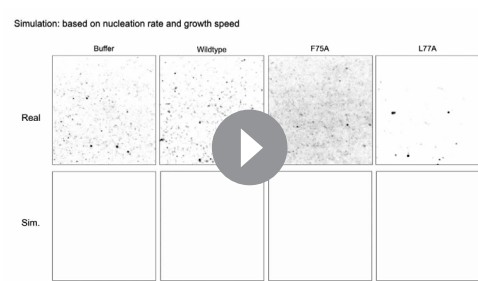

**Video 3.** Video of simulated γTuNA-dependent activation of γTuRC.

https://elifesciences.org/articles/80053/figures#video3

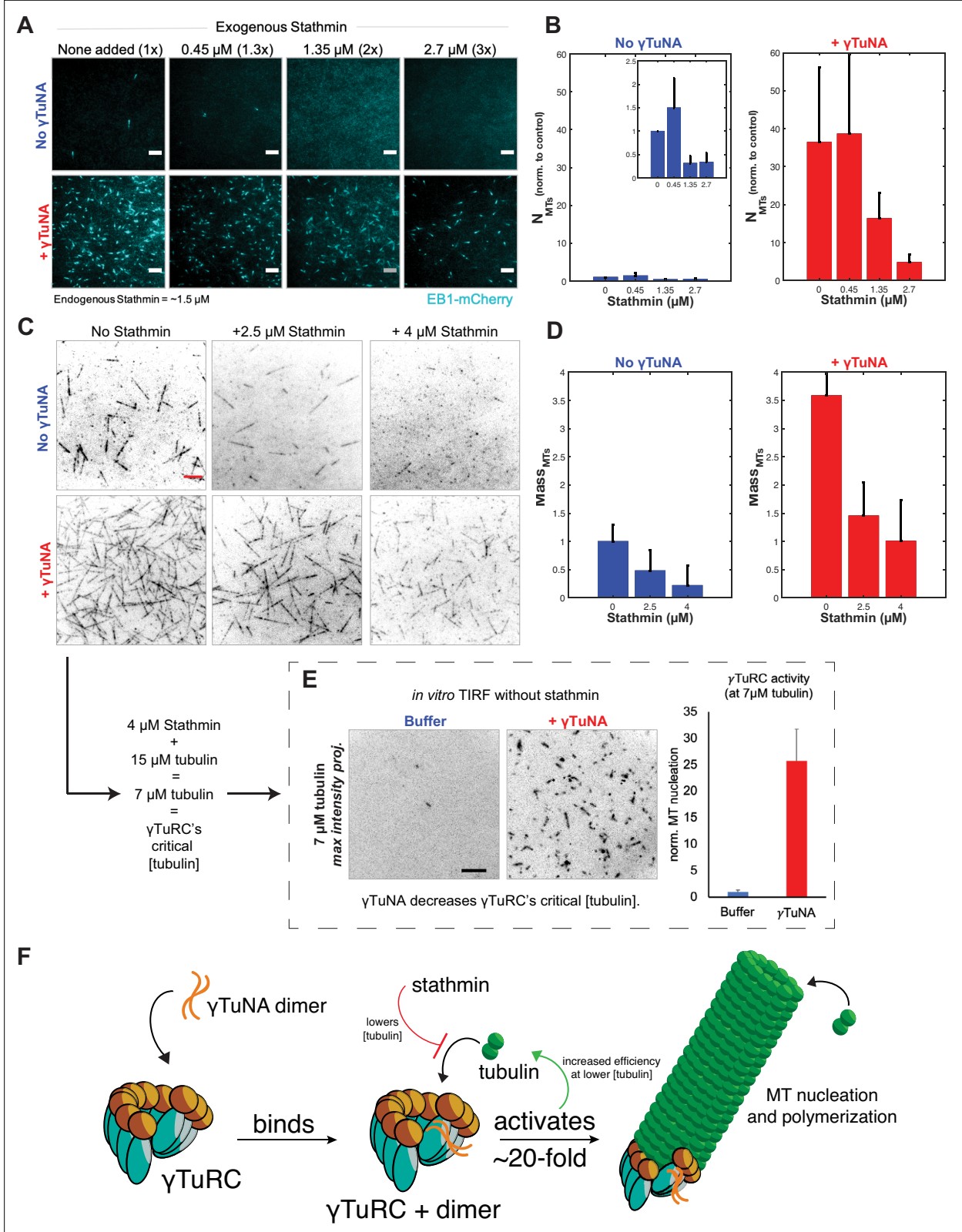

**Figure 6.** γTuNA enhances γTuRC activity at low tubulin concentrations in extract and in vitro. (**A**) TIRF assay of extract MT nucleation (Strep-His-γTuNA vs stathmin). His-SNAP-tag-*Xenopus* stathmin (isoform 1 A) was added at 0.45, 1.35, and 2.7 µM final concentration to extract. Either buffer or 2.3 µM (1.15 µM dimer) Strep-His-*Xenopus* wildtype γTuNA was also added. EB1-mCherry was used to visualize MT plus-ends (MT number). Images were taken after 5 min at 18–20°C and pseudo-colored cyan. Bars = 5 µm. (**B**) Normalized MT number for each concentration of stathmin tested, n=4; bars

*Figure 6 continued on next page*

*Figure 6 continued*

are SEM. (**C**) Single molecule TIRF assay of γTuRC MT nucleation in vitro. Purified γTuRCs were assayed at 33.5 °C with 15 μM tubulin, 1 mM GTP, either alone, in the presence of 2.5 μM or 4 μM stathmin, or with 3.3 μM (1.65 μM dimer) Strep-His-*Xenopus* γTuNA. 5% Alexa-568-tubulin was used to visualize MTs. Red bar = 5 μm. (**D**) Normalized MT mass at 300 seconds (intensity normalized to buffer control) from C. Error bars are SEM. N=3 minimum for all conditions, except: n=4 for "4 μM stathmin" and, n=5 for "4 μM stathmin + γTuNA". (**E**) TIRF assay of γTuRC activity in vitro at its critical tubulin concentration (7 μM) with or without 3.3 μM (1.65 μM dimer) Strep-His-*Xenopus* γTuNA. Images are max intensity projections of 5 min time-series. Normalized to buffer control: n=3 (γTuNA); n=2 (buffer). (**F**) Model of γTuNA's activation of γTuRC-mediated microtubule nucleation. See "*Figure 6— source data 1*" for all numerical data.

The online version of this article includes the following source data for figure 6:

**Source data 1.** Numerical data used in *Figure 6*: including raw EB1 counts for stathmin/γTuNA experiments in extract, and MT mass measurement for in vitro TIRF assays with purified γTuRC, γTuNA, and stathmin.

MT mass (*Figure 6C–D*, '+ γTuNA'). In fact, at 4 μM stathmin with γTuNA, MT mass levels were restored to the level of the γTuRC control (no stathmin, no γTuNA). This suggested that γTuNA indirectly counteracts the effect of stathmin in extract by increasing the efficiency of γTuRC-mediated MT nucleation at lower tubulin concentrations.

To test this possibility, we assayed γTuRC activity in vitro at its previously reported critical concentration of 7 μM tubulin; at or below this concentration γTuRC activity is minimal to non-existent (*Consolati et al., 2020*; *Thawani et al., 2020*). This tubulin concentration is also equivalent to that in our assay in *Figure 6C*, as 4 μM stathmin will remove 8 μM tubulin, leaving only 7 μM tubulin free in a 15 μM reaction. As expected, at this critical concentration γTuRC had little detectable activity with only ~1–2 MTs nucleated over a five-minute time course (*Figure 6E*, buffer). By contrast, γTuRC activity increased twenty-five-fold when in the presence of γTuNA (*Figure 6E*). Thus, γTuNA decreases γTuRC's critical tubulin concentration, or restated, increases its ability to nucleate MTs at lower tubulin concentrations. We believe this result explains how γTuNA indirectly counteracts the inhibitory effect of stathmin on γTuRC.

## Discussion
### A model for γTuNA-mediated activation of γTuRC

In this work, we investigated how MT nucleation is regulated by Cdk5rap2's γTuNA-mediated activation of γTuRC (*Figure 6F*). We showed that the γTuNA domain is an obligate dimer and that both dimerization and the F75 residue are crucial for binding γTuRC, providing the first biochemical validation for a recent structure of γTuRC bound to a putative γTuNA dimer (*Wieczorek et al., 2020a*). Moreover, we defined other core residues required for both γTuNA dimerization and subsequent γTuRC binding. We found that γTuNA dimers directly activate γTuRC-dependent MT nucleation in extract and in vitro. Finally, we uncovered that γTuNA dimers overcome barriers to MT nucleation posed by the tubulin sequestrator stathmin by enhancing γTuRC's activity at low tubulin concentrations. Because γTuNA domains are also found in myomegalin and the branching factor TPX2, among others, our findings are broadly applicable to multiple MTOCs and model eukaryotes from yeast to humans.

It remains an open question how the γTuNA dimer directly enhances γTuRC's nucleation activity. It is tempting to speculate that γTuNA binding triggers a conformational change of γTuRC from its wide diameter lattice to a closed state (*Liu et al., 2020*; *Wieczorek et al., 2020a*; *Consolati et al., 2020*). Because previous work has characterized the structures of γTuNA bound to human and *Xenopus* γTuRC and observed no obvious structural change (*Liu et al., 2020*; *Wieczorek et al., 2020a*; *Wieczorek et al., 2020b*), it is possible that binding a γTuNA dimer only transiently biases γTuRC toward the closed ring conformation. Furthermore, our own prior modeling suggested that the free energy provided by stochastic binding and lateral association of the first 3–4 tubulin dimers is sufficient to overcome the energy barrier between γTuRC's open and closed states (*Thawani et al., 2020*). The γTuNA domain possibly lowers this energy barrier for ring closure, as evidenced by its ability to lower γTuRC's critical tubulin concentration. Ultimately, detection of an activated state of vertebrate γTuRC may require the presence of tubulin, GTP, and the γTuNA domain combined with sophisticated structural methods.

Intriguingly, recent work in yeast suggests that near-closure, or biasing, of the γTuRC ring might require simultaneous binding by multiple CM1/γTuNA domains present in proteins like Spc110p (*Brilot et al., 2021*). Unlike in vertebrate structures of γTuRC, yeast γTuRC appears to have six well-defined CM1/γTuNA binding sites, where overhanging CM1 domains facilitate the formation of the ring by reinforcing lateral interactions between its constituent γ-tubulin small complex (γTuSC) subunits (*Brilot et al., 2021*). This might suggest that additional γTuNA binding sites may also be present in vertebrate γTuRCs, and simultaneous binding of multiple γTuNAs could further enhance γTuRC activity. As of yet, any additional γTuNA binding sites have not been detected in vertebrate γTuRC structures, but this possibility is exciting and can actively be investigated with single molecule imaging of fluorescently labeled γTuRC and γTuNA.

Altogether, our current model for γTuNA activation of γTuRC involves a direct binding event between a dimerized γTuNA-containing protein (e.g. Cdk5rap2) and γTuRC, which activates γTuRC's MT nucleation ability ~20-fold. This activation overcomes negative regulation by stathmin in the cytoplasm or low local tubulin concentrations (*Figure 6F*). Once bound, these activated γTuRCs are enriched by specific MTOCs, like the centrosome, via additional localization motifs present in the γTuNA-containing protein. As Cdk5rap2 is also present in the cytoplasm it may be possible that some level of activation also occurs outside MTOCs, although recent work hints at how both access to the γTuNA domain and Cdk5rap2 localization is regulated via phosphorylation to prevent ectopic activation (*Conduit et al., 2010*; *Conduit et al., 2014*; *Hanafusa et al., 2015*; *Feng et al., 2017*; *Tovey et al., 2021*).

We note that the γTuNA domain's enhancement of γTuRC activity may be greater than our reported 20-fold. Based on our analysis, an estimated 33% of our purified γTuRCs likely retain a γTuNA dimer at the end of the purification (*Figure 2—figure supplement 2C*). This means that our baseline ('buffer') level of γTuRC activity is likely higher than might otherwise be observed with a γTuNA-independent purification. We do not believe this slightly elevated baseline level interferes with demonstrating that γTuNA dimers activate γTuRC, but rather stress that γTuNA dimers might have an even more potent effect on γTuRC activity than we report here.

## Divergent behavior among γTuNA mutants is reflective of aspects critical for the γTuRC interaction

In the experiments presented in *Figures 3 and 4*, we found that the L70A and L77A mutants similarly formed intermediate SEC peaks between full dimer and monomer (*Figure 3C*) but had divergent γTuRC binding ability in extract pulldowns (*Figure 3E*). This suggests that the region of the γTuNA coil from position 75 to position 77 (F75, L77) is the core γTuNA-γTuRC binding interface, within which mutations are not well-tolerated for stable γTuRC interaction in extract. We observed that mutations at positions moving from this core towards the N-terminus (L70 to I67 to F63) had less and less impact on both dimerization and γTuRC binding ability in extract (*Figure 3*). Thus, the divergent behavior for L70A and L77A appears to be a result of L70's position outside the most critical region, retaining a small amount of γTuRC binding. However, as our extract assays demonstrate in *Figure 4*, this small amount of binding by L70A is not sufficient to significantly activate γTuRC in extract.

Interestingly, we also found that the double I67A/L70A mutant had a strong loss of dimerization but still retained some γTuRC binding, just below the level of the L70A single point mutant (*Figure 3E and G*). By comparing this to the human I67D/L70D mutant (*Figure 3F*) we found that double substitution to aspartate, instead of alanine, completely removed this residual γTuRC binding. This suggests that retaining some hydrophobicity at these positions might preserve enough of the coil structure to allow for a weak interaction with γTuRC, despite lacking the required hydrophobicity to form a stable coiled-coil dimer (*Figure 3A*). In support of this, closer inspection of the peak SEC retention volumes (*Figure 3B–C*) reveals that human I67D/L70D is eluted ahead of human I67A/L70A (~15.0 vs ~15.2 mL), indicating that I67D/L70D has a larger hydrodynamic radius despite differing in only two residues. We believe this difference is reflective of changes in the γTuNA coil structure, where the hydrophilic aspartate residues now cause the coil to extend, kink, or otherwise deform in a way that increases the hydrodynamic radius of the I67D/L70D protein. This drastic change in the local coil structure, in addition to blocking coiled-coil dimerization, also likely prevents even weak interactions with γTuRC.

Yet, I67A/L70A still retains a small amount of residual binding to γTuRC. How might this I67A/L70A mutation be overcoming the loss of dimerization to weakly bind γTuRC? It is possible that two

separate monomeric coils of mutant γTuNA might bind the same γTuRC and form a weak complex. In this scenario, the interaction with the γTuRC would stabilize the γTuNA dimer, overcoming the loss of the strongly hydrophobic contacts normally present in the coiled-coil dimer interface. We believe that we have observed a related phenomenon with the *Xenopus* L77A mutant in our in vitro reactions (*Figure 5B*), where late in the assay L77A can begin to increase γTuRC activity despite lacking strong dimerization and strong γTuRC binding ability (*Figure 3B–E*). We hypothesize that this late effect is reflecting mutant L77A monomers that are stochastically stabilized into a dimer on γTuRC (*Figure 5— figure supplement 3*). We further predict that this is also a function of the in vitro environment, which is more permissive of these types of interactions, as L77A does not display this behavior in extract. The fact that I67A/L70A can weakly bind in extract is likely due to the fact that I67 and L70 are outside the critical core region discussed above (aa 75–77). Furthermore, we predict that this weak binding can only occur in hydrophobic-to-weaker-hydrophobic substituted versions of γTuNA, like I67A/L70A or L77A. These types of substitutions are not as likely to cause drastic changes to the overall coil structure of a γTuNA monomer, which might allow for two of these monomers to be stabilized into a dimer on γTuRC.

Finally, our γTuNA-GCN4 fusion constructs were our attempt to rescue the coiled-coil dimer and subsequent γTuRC binding ability (*Figure 3—figure supplement 1*). While this did rescue dimerization in an SEC assay (*Figure 3—figure supplement 1*), these fusion constructs did not rescue γTuRC binding. This divergent result is likely because the fused GCN4 domain did not restore the local coil structure of γTuNA (if impacted). Our GCN4 fusion also has no impact on the hydrophobic character of the core region (aa 75–77) an aspect which appears to be most critical for γTuRC binding in extract. Also, we suggest that specific residues might be required for both dimerization and for making specific contacts with γTuRC. For these cases, inducing dimerization would never be sufficient to restore wildtype levels of γTuRC binding as the specific residue enabling stable interaction would still be missing. We imagine that the core residues, like L77, have twin impacts on both dimerization and stable γTuRC binding.

We propose that dimerization is a key component of how γTuNA interacts with γTuRC (supported by the cryo-EM structure by Wieczorek et al., Cell Reports, 2020), but dimerization on its own is not sufficient. Indeed, the F75 residue, which would be located on the outer surface of the dimer (*Figure 3A*), was required for activation and strong binding in all our assays. We hypothesize that this is likely due to a stabilizing or docking role where this outer surface residue helps 'lock' the γTuNA domain into γTuRC.

## Resolving conflicting data concerning γTuNA's effect on γTuRC

This study was partly motivated by an apparent discrepancy between the original reports of γTuNA's ability to activate γTuRC in vivo and in vitro (*Fong et al., 2008*; *Choi et al., 2010*; *Muroyama et al., 2016*) and more recent in vitro data from our group and others that found little to no effect (*Liu et al., 2020*; *Thawani et al., 2020*).

In their recent structural study of antibody-purified *Xenopus* γTuRC, Liu and colleagues concluded that the N-terminal region of Cdk5rap2 (or CEP215) containing the γTuNA domain had little to no effect on γTuRC MT nucleation in vitro (Extended Data Fig. 9b from *Liu et al., 2020*). They did, however, report that wildtype γTuNA (CEP215N) co-precipitated γ-tubulin, while the F75A mutant did not (Extended Data Fig. 9c from *Liu et al., 2020*). Liu and colleagues used N-terminally GST tagged versions of γTuNA (CEP215N). Like our colleagues, we find that the presence of a large N-terminal tag does not interfere with γTuNA's ability to bind γTuRC (*Figure 3*). However, our studies revealed that a bulky N-terminal Halo-tag on γTuNA turns this activator into a specific inhibitor of γTuRC-mediated MT nucleation in extract and in vitro (*Figure 5—figure supplement 2*). This is likely due to the steric clash produced by two copies of the bulky N-terminal tag in proximity to the critical nucleation interface on the γ-tubulin ring.

Slightly confounding, in a previous fixed in vitro assay (*Muroyama et al., 2016*) Muroyama and colleagues did observe an activation effect with an N-terminally GST-tagged truncation of CDK5RAP2. This suggests that differences in the distance between the bulky tag and γTuNA, as well as the ratio of γTuNA to γTuRC tested, determines whether an activation effect is possible. If it is true that multiple γTuNA binding sites exist in vertebrate γTuRCs (as in yeast; *Brilot et al., 2021*), then this N-terminal tag effect might be further compounded as multiple steric clashes could be present. We note that the

original reports from the Qi group used the small FLAG tag (*Choi et al., 2010*), and our work is based on the small Strep-His tag at the N-terminus.

Finally, in our prior work (*Thawani et al., 2020*), we had established an antibody-based *Xenopus* γTuRC purification, albeit with limited yield and batch-to-batch variability. Although N-terminally 6xHis-tagged γTuNA activates MT nucleation in extract (as presented in *Figure 1* of this study), it had little to no effect on the original antibody-purified *Xenopus* γTuRC in vitro (Figure 6 in *Thawani et al., 2020*). This inability to activate antibody purified γTuRC puzzled us. We initially thought that an additional factor might be required for γTuNA-mediated activation of γTuRC. However, even with mass spectrometry data from our group and others (*Liu et al., 2020*; *Consolati et al., 2020*; *Wieczorek et al., 2020a*), we did not find an obvious target. Since then, we developed the Halo-γTuNA purification method described here, which is routinely at least 20-fold higher yield, higher purity, and ultimately has more robust activity. This resulted in increased density of nucleation competent γTuRCs present in our single molecule assays, as well as better detection of γTuNA's activation effect. Silver staining the peak γTuRC fraction for our new prep (*Figure 2—figure supplement 2B*) showed the same banding pattern as that published with our previous antibody prepped γTuRC, indicating that aside from γTuRC components, there was no obvious major factor present to explain the response to γTuNA. Rather, we believe the difference can be explained by the greater yield and consistent quality of γTuRC provided by the Halo-γTuNA prep. As such, we validate and extend the original γTuNA studies by the Qi group.

## Ideas and speculation

### Other factors possibly involved in tuning $\gamma$ TuNA-$\gamma$ TuRC activity

Our mass spectrometry analysis revealed that γTuRC is the dominant co-precipitant for wildtype versions of human and *Xenopus* γTuNA (*Figure 2—figure supplement 1*). We also detected the nucleoside diphosphate kinase 7 (better known as NME7). This agrees with prior work showing that NME7 is a γTuRC subunit that is present regardless of how γTuRC is purified or whether γTuNA is present (Hutchins et al., Science, 2010; Teixido-Traversa et al., Mol Biol Cell, 2010; *Liu et al., 2014*; *Liu et al., 2020*; *Consolati et al., 2020*; *Wieczorek et al., 2020a*). However, how NME7 contributes to γTuRC's activity, or its regulation is unknown.

Surprisingly, we detected three unique proteins that were enriched at a higher level than NME7 (*Figure 2—figure supplement 1*) and had not been reported to directly interact with γTuRC or γTuNA. These were the cyclin-dependent kinase 1 (CDK1) subunits A and B, as well as the type II delta chain of the calmodulin-dependent protein kinase (CAMK2D). Hence, these might be novel co-factors for γTuRC.

While our work has now revealed that γTuNA-containing proteins can directly activate the MT nucleation template, γTuRC, several questions remain. Chief among these is whether the co-nucleation factor, XMAP215/ch-TOG, which is now known to act with γTuRC to nucleate MTs (*Thawani et al., 2018*), might further enhance γTuNA's effect on γTuRC. Or in a similar vein, what effect does the aforementioned γTuRC subunit NME7 have on γTuNA-triggered activation? Finally, we are excited by the possibility that a γTuNA-bound γTuRC might form a novel interface recognized by other factors. Investigating this novel interface and how multiple factors simultaneously tune γTuRC activity is an exciting avenue that can further our understanding of microtubule nucleation.

## Materials and methods

### Key resources table

| Reagent type (species) or resource | Designation | Source or reference | Identifiers | Additional information |
|---|---|---|---|---|
| Gene (*Xenopus laevis*) | cdk5rap2.L (*Xenopus laevis*) | NCBI | XP_018085184.1; isoform X1 | |
| Strain, strain background (*Escherichia coli*) | DH5-alpha (High Efficiency) | New England Biolabs | C2987I | Chemically competent; cloning strain |

*Continued on next page*

*Continued*

| Reagent type (species) or resource | Designation | Source or reference | Identifiers | Additional information |
|---|---|---|---|---|
| Strain, strain background (*Escherichia coli*) | BL21(DE3) | Sigma-Aldrich | 71402 | Chemically competent; expression strain |
| Biological sample (*Xenopus laevis*) | *Xenopus laevis* eggs and egg extract | This study | | Method previously described; see **Good and Heald, 2018** |
| Antibody | anti-MZT1, (Rabbit polyclonal) | Abcam | ab178359 | 375 µg/mL, used in bead attachment assay |
| Antibody | Anti-gamma-tubulin, GTU-88 clone (Mouse, monoclonal) | Sigma | T6557; RRID:AB_2863751 | 1:1000 dilution |
| Antibody | anti-GCP5, E-1 clone (Mouse monoclonal) | Santa Cruz Biotechnology | sc-365837; RRID:AB_10847352 | 1:250 dilution |
| Antibody | anti-GFP, ChIP grade, (Rabbit polyclonal) | Abcam | Ab290 | 1:1000 dilution |
| Antibody | anti-StrepTagII, (Mouse monoclonal) | Qiagen | 34850; RRID:AB_2810987 | 1:1000 dilution |
| Antibody | anti-AU1, (Mouse monoclonal) | Biolegend | 901905 | 1:1000 dilution |
| Antibody | Mouse IgG, HRP-linked whole Ab, secondary (Sheep, clonality not reported by manufacturer) | Amersham | NA931-1ML | 1:3000 dilution |
| Antibody | Rabbit IgG, HRP-linked whole Ab, secondary (Donkey, clonality not reported by manufacturer) | Amersham | NA934-1ML | 1:3000 dilution |
| Recombinant DNA reagent | pET28a-Hook3 aa 1–160-GCN4 (plasmid) | Addgene | 74608; RRID:Addgene_74608 | Ron Vale; **Schroeder and Vale, 2016** |
| Recombinant DNA reagent | Modified pST50Trc (StrepTagII-6xHis-PreScission cleavage site) with human or *X. laevis* γTuNA for bacterial expression (plasmids) | This study | | See *Table 1* for all constructs |
| Commercial assay or kit | 2 x Gibson Assembly Master Mix | New England Biolabs | E2611L | |
| Commercial assay or kit | Strep-Tactin Superflow | IBA | 2-1206-025 | |
| Commercial assay or kit | Halo Magne beads | Promega | G7287 | |
| Commercial assay or kit | HisPur Ni-NTA magnetic beads | ThermoFisher | 88831 | |
| Commercial assay or kit | Quick Start Bradford 1 x Dye Reagent | BioRad | 5000205 | |
| Commercial assay or kit | Akta Püre System with Superdex 200 increase 10/300 GL column | Cytiva (formerly GE Healthcare) | 28-9909-44 | |
| Commercial assay or kit | SNAP i.d. 2.0 rapid Western blotting system | EMD-Millipore | SNAP2MM | |
| Chemical compound, drug | Pluronic-F127 | ThermoFisher | P6866 | |
| Chemical compound, drug | NHS-PEG4-Biotin | ThermoFisher | A39259 | |
| Chemical compound, drug | Dichlorodimethylsilane | Sigma | 440272–100 ML | |
| Software, algorithm | Fiji (ImageJ) | NIH | RRID:SCR_002285 | |
| Software, algorithm | MATLAB | MathWorks | ver. R2019a; RRID:SCR_001622 | |

*Continued on next page*

*Continued*

| Reagent type (species) or resource | Designation | Source or reference | Identifiers | Additional information |
|---|---|---|---|---|
| Software, algorithm | Prism 7 | GraphPad Software | RRID:SCR_002798 | |
| Software, algorithm | Relion 3.1 | *Zivanov et al., 2018* | | |
| Software, algorithm | CryoSparc 3.2 | *Punjani et al., 2017* | | |
| Other | Nikon Ti-E inverted scope system | Nikon | RRID:SCR_021242 | See Materials and Methods. |
| Other | Optima MAX-XP ultracentrifuge | Beckman Coulter | 393315 | See Materials and Methods. |

## Cloning and purification of human and *Xenopus* γTuNA constructs

The fragment of human CDK5RAP2 (51-100) containing the CM1 motif/γTuNA domain was sub-cloned into a bacterial expression pST50 vector using Gibson cloning. This vector was engineered with N-terminal Strep-TagII, 6xHis, TEV cleavage, HaloTag, and PreScission 3 C protease cleavage sites. This vector was then truncated to human CDK5RAP2 aa 53–98 (see *Table 1*). The resulting construct, Strep-His-TEV-HaloTag-3C-human γTuNA, (Halo-human γTuNA), was expressed in Rosetta 2 (DE3) *E. coli* cells. Rosetta 2 cells were grown in 2 L terrific broth (TB) cultures to O.D.=0.7 and induced with 0.5 mM IPTG at 16 °C for 18 h. The cultures were pelleted, snap-frozen, and stored at –80° C.

The CM1/γTuNA-motif in *Xenopus laevis* Cdk5rap2 was confirmed via sequence alignment to the human version (*Figure 1B*) and inserted using Gibson cloning into the same pST50 bacterial construct as above. This generated Strep-His-TEV-HaloTag-3C-*Xenopus* γTuNA (Halo-*Xenopus* γTuNA; *Xenopus* Cdk5rap2 aa 56–101). The loss-of-function mutation, F75A, first identified by *Fong et al., 2008*, was introduced into the Halo-*Xenopus* γTuNA sequence at the equivalent, conserved phenylalanine at position 71 to make Strep-His-TEV-HaloTag-3C-*Xenopus* γTuNA "F75A" (Halo-*Xenopus* γTuNA F75A). Both these constructs were expressed as described above with the human version.

The amino acid sequence of the human γTuNA used is: SPTRARNMKDFENQITELKKENFNLKLR IYFLEERMQQEFHGPTEH. The sequence for the *Xenopus* γTuNA (wildtype) used in this work is: MKDF EKQIAELKKENFNLKLRIYFLEEQVQQKCDNSSEDLYRMNIE. All γTuNA constructs generated in this study are listed in *Table 1*. For GCN4 C-terminal fusions, we used pET28a-Hook3 aa 1–160-GCN4 plasmid, which was a gift from Dr. Ron Vale (Addgene plasmid # 74608; RRID:Addgene_74608).

To purify the human and *Xenopus* Halo-γTuNA constructs, 2 L TB cell pellets were thawed on ice and resuspended into 50 mL of Strep Lysis Buffer (50 mM TRIS, pH = 7.47, 300 mM NaCl, 6 mM β-mercaptoethanol, 200 μM PMSF, 10 μg/mL DNase I), and a single dissolved cOmplete EDTA-free Protease Inhibitor Cocktail tablet (cat # 11873580001, Roche). Cells were resuspended using a Biospec Tissue Tearor (Dremel, Racine, WI) and lysed in an Emulsiflex C3 (Avestin, Ottawa, Canada) by processing four times at 10,000–15,000 psi. Cell lysate was spun at 30,000 rpm for 30 min, 2 °C in a Beckman Optima-XE 100 ultracentrifuge, 45Ti rotor. Supernatant was then passed twice through a 15 mL column volume (CV) of Strep-Tactin Superflow resin (IBA, Goettingen, Germany). The column was then washed with 10 CV of Strep Bind buffer (50 mM TRIS, pH = 7.47, 300 mM NaCl, 6 mM β-mercaptoethanol (BME), 200 μM PMSF). The γTuNA proteins were then eluted with 1.5 CV of Strep Elution buffer (Strep Bind Buffer with 3.3 mM D-desthiobiotin (cat. #2-1000-005, IBA)). Yield and purity were assessed via SDS-PAGE gel and Coomassie stain. Concentration was assessed via Bradford assay. All γTuNA constructs yielded between 40 and 60 mg of protein (per 2 L TB culture) at >98% purity.

## Size-exclusion chromatography of Halo-γTuNA proteins

For all size-exclusion assays, we used an ÄKTA Pure system with a Superdex 200 increase 10/300 GL column (cat. #2, Cytiva, Marlborough, MA), with a 500 μL manual injection loop. All assays were done in CSFxB, 6 mM BME, without sucrose at 4 °C. Strep-His-TEV-Halo-3C-γTuNA constructs shown in *Figure 3* were run at ~50 μM final concentration in a total volume of 550 μL Strep bind buffer (see above), at a 0.7 mL/min flow rate. Absorbance at 280 nm was used to track the protein peak. Each trace was normalized by the maximum peak for that run, prior to combined plotting with all other constructs in MATLAB (ver. R2019a, MathWorks, Natick, MA; RRID:SCR_001622).

**Table 1.** γ TuNA constructs generated in this study.
Key: S = StrepTagII, H = 6xHis-tag, 3C = human rhinovirus 3C (PreScission) protease cleavage site, TEV = tobacco etch virus protease cleavage site, GCN4 = yeast transcription factor GCN4 dimerization domain. Mutated residues in the "Sequence" column are designated using { } brackets. Primer sequences and a copy of this table are included in ***Supplementary file 1***.

| Name | Species | N-term tag versions | Fused GCN4 version? | Sequence | Mutation Type |
|---|---|---|---|---|---|
| *Wildtype* | H. sapiens (H.s) | SH-3C; SH-Halo-3C; SH-Halo-3C-AU1 | No | SPTRARNMKDFENQITELKKENFNLKLRIYFLEERMQQEFHGPTEH | None |
| *Wildtype* | X. laevis (X.l.) | SH-TEV; SH-3C; SH-TEV-GFP; SH-Halo-3C- | Yes, SH-Halo-3C-(γTuNA)-GCN4 | ...MKDFEKQIAELKKENFNLKLRIYFLEEQVQQKCDNSSEDLYRMNIE | None |
| *F63A* | H.s./X.l. | SH-3C; SH-Halo-3C | No | ...MKD(A)EKQIAELKKENFNLKLRI... | Weakened hydrophobic |
| *I67A* | H.s./X.l. | SH-3C; SH-Halo-3C | Yes, SH-Halo-3C-(γTuNA)-GCN4 | ...MKDFEKQ(A)AELKKENFNLKLR... | Weakened hydrophobic |
| *L70A* | H.s./X.l. | SH-3C; SH-Halo-3C | Yes, SH-Halo-3C-(γTuNA)-GCN4 | ...MKDFEKQIAE(A)KKENFNLKLR... | Weakened hydrophobic |
| *F75A* | H.s./X.l. | SH-3C; SH-Halo-3C | Yes, SH-Halo-3C-(γTuNA)-GCN4 | ...MKDFEKQIAELKKEN(A)NLKLR... | Weakened hydrophobic |
| *L77A* | H.s./X.l. | SH-3C; SH-Halo-3C | Yes, SH-Halo-3C-(γTuNA)-GCN4 | ...MKDFEKQIAELKKENFN(A)KLR... | Weakened hydrophobic |
| *I67A/L70A* | H.s./X.l. | SH-3C; SH-Halo-3C | No | ...MKDFEKQ(A)AE(A)KKENFNLKLR... | Weakened hydrophobic (2 x) |
| *F63D* | X.l. | SH-Halo-3C | No | ...MKD(D)EKQIAELKKENFNLKLR... | Flip to hydrophilic |
| *I67D* | X.l. | SH-Halo-3C | No | ...MKDFEKQ(D)AELKKENFNLKLR... | Flip to hydrophilic |
| *L70D* | X.l. | SH-Halo-3C | No | ...MKDFEKQIAE(D)KKENFNLKLR... | Flip to hydrophilic |
| *F75D* | X.l. | SH-Halo-3C | No | ...MKDFEKQIAELKKEN(D)NLKLR... | Flip to hydrophilic |
| *L77D* | X.l. | SH-Halo-3C | No | ...MKDFEKQIAELKKENFN(D)KLR... | Flip to hydrophilic |
| *I67D/L70D* | H.s./X.l. | SH-Halo-3C | Yes, SH-Halo-3C-(γTuNA)-GCN4 | ...MKDFEKQ(D)AE(D)KKENFNLKLR... | Flip to hydrophilic (2 x) |
| *I67D/L70D/L77D* | H.s. | SH-Halo-3C | No | SPTRARNMKDFENQ(D)TE(D)KKENFN(D)KLRIYFLEERMQQEFHGPTEH | Flip to hydrophilic (3 x) |

## TIRF imaging of MT nucleation in *Xenopus laevis* egg extracts

*Xenopus laevis* egg extracts were prepared as previously described (*Good and Heald, 2018*). For assaying γTuNA's effect on MT nucleation levels, 7.5 µL of extract were incubated on ice with 0.5 µL 10 mM Vanadate (0.5 mM final), 0.5 µL 1 mg/mL end-binding 1 (EB1)-mCherry protein (0.05 mg/mL final), 0.5 µL of 1 mg/mL Cy5-tubulin (0.05 mg/mL final), 0.5 µL of CSFxB (10% sucrose), and 0.5 µL of TRIS control buffer (50 mM TRIS, pH = 7.47, 300 mM NaCl, 6 mM β-mercaptoethanol) or 0.5 µL of γTuNA protein (previously diluted in Tris control buffer such that final concentrations are as stated in *Figure 1*). Reactions (10 µL total) were gently mixed by pipetting once, before adding to a channel on a 6-channel slide at 18–20°C. All γTuNA concentrations for each condition (wildtype or F75A mutant) were imaged in parallel on the same slide. Multi-channel images were acquired sequentially and at 1 min intervals using the NIS-Elements AR program (NIKON, ver. 5.02.01-Build 1270; RRID:SCR_014329). The 647 nm/Cy5 channel (excitation: 678 nm, emission: 694 nm) was used for microtubules (MT) and the 561 nm channel for EB1 MT plus-tips (ex: 587 nm, em: 610 nm). The images were captured on a Nikon Ti-E inverted system (RRID:SCR_021242), with an Apo TIRF 100 x oil objective (NA = 1.49), and an Andor Neo Zyla (VSC-04209) camera with no binning and 100 ms exposures. Resulting images were 2048 by 2048 pixels (132.48 µm x 132.48 µm). The 561 nm channel was pseudo-colored green.

To quantify MT nucleation levels, we extracted the number of MT plus ends (tracked by EB1 spots) for each condition using the 561 nm/EB1-mCherry channel. We wrote a macro in FIJI (ImageJ; *Schindelin et al., 2012*; RRID:SCR_002285) to automate counting EB1 spots. Briefly, 50 µm by 50 µm (800x800 pixels$^2$) representative windows were cropped from each field of view, smoothed using the FIJI function ('Process—>Smooth'), and thresholded using the Yen option. The built-in FIJI functions 'Find 'dges" and 'Analyze particles' were then used to count the number of thresholded spots. These values, representing the number of MT plus ends, were then normalized to the buffer control for each condition to obtain the fold change in MT number. These fold changes were then averaged across four biological replicates (independent extract preps) and plotted using Prism GraphPad 7 (GraphPad Software, San Diego, CA; RRID:SCR_002798). Representative images are shown in *Figure 1*. The 95% confidence intervals and SEM are also shown.

## γTuNA and Stathmin in Extract TIRF assays

For assaying γTuNA dimer mutants' effects on MT nucleation, we used the above procedure except all constructs shown in *Figure 4* were tested at 2 µM final concentration using 600 msec exposures of EB1 (561 nm channel) only. For assaying γTuNA's effect on stathmin in extract (*Figure 6*), we used the same procedure above except we added either 0.5 µL unlabeled 6xHisTag-SNAP-Stathmin or Tris control buffer, instead of 0.5 µL CSFxB. For *Figure 4*, the number of EB1 spots were normalized by the buffer reaction, followed by normalization by the wildtype γTuNA reaction. Data were then averaged and plotted in MATLAB. For *Figure 6*, data were normalized to the buffer reaction, averaged, and the plotted in MATLAB. For both assays, bars are means and error bars are SEM.

## TIRF imaging of MT nucleation from γTuNA-coated beads in vitro

### γ TuNA bead assay (endpoint version)

We first saturated 5 µL of micron-scale, HisPur Ni-NTA magnetic beads (cat. # 88831, ThermoFisher) with 50 µL of either bovine serum albumin (6.5 mg/mL BSA, as mock), wildtype *Xenopus* Strep-His-γTuNA (71 µM), or *Xenopus* Strep-His-γTuNA F75A mutant (~117 µM) in CSFxB buffer. After 30 min incubation on ice, beads were removed with a magnet, washed with 150 µL CSFxB, and resuspended with 50 µL BRB80 buffer (80 mM PIPES, pH = 6.8 with KOH, 1 mM MgCl$_2$, 1 mM EGTA). These beads were then diluted 1/1000 in polymerization mix (15 µM total tubulin with 5% labeled Cy5-tubulin and 1 mM GTP in BRB80 buffer) and added to a channel on a glass slide. Beads were located via differential interference contrast (DIC) microscopy. Then MT aster formation for each condition was imaged via oblique TIRF microscopy at 5 min intervals up to 25 min.

### Anti-Mzt1 γTuNA bead assay (live imaging version)

We first passivated glass coverslips with dichlorodimethylsilane (DDS, cat. #440272–100 ML, Sigma), as previously published (*Gell et al., 2010*; *Alfaro-Aco et al., 2017*). These coverslips were then attached

with double-sided tape to glass slides to create multi-channel imaging chambers. To each chamber, we added in order: (1) 50 µL of BRB80, (2) 20 µL of 375 µg/mL Mzt1 antibody in BRB80 (anti-Mzt1, cat. # ab178359, Abcam), (3) 1% Pluronic-127 (cat. # P6866, ThermoFisher) in BRB80, and (4) 10 µL of 1/1000 diluted wildtype γTuNA beads in BRB80 (pulled from extract). At each step, we paused for 5 min incubations at room temperature. Just prior to imaging via TIRF, we then added 10 µL of ice-cold BRB80, followed by cold polymerization mix (15 µM total tubulin with 5% labeled Cy5-tubulin and 1 mM GTP in BRB80). Images were taken at room-temperature (18–20°C).

## Purification of native *Xenopus* γ-tubulin ring complex (γTuRC) via Halo-human γTuNA pulldown

To purify native *Xenopus* γTuRC from *Xenopus* egg extract, we employed a strategy similar to previous work (*Wieczorek et al., 2020b*) and originally observed by *Choi et al., 2010*. Here we similarly use γTuNA as a bait for γTuRC, except we use the human version of Halo-γTuNA directly coupled to beads via the affinity of the Halo-tag for its substrate. These beads are then used to pulldown γTuRC from *Xenopus laevis* egg extract.

*Xenopus laevis* egg extracts were prepared as described previously. Extracts were snap-frozen in liquid nitrogen and stored at –80 °C. One day prior to purification, 15 mg of Halo-human γTuNA were thawed and diluted to ~1 mg/mL in a 15 mL conical tube with Coupling Buffer (20 mM HEPES, pH = 7.5 with KOH, 75 mM NaCl) to a final NaCl concentration of 100–135 mM NaCl. Next, 2 mL of Halo Magne bead slurry (cat. #G7287, Promega, Madison, WI) were washed with MilliQ water and 3 CV of modified CSF-XB, 2% sucrose buffer (10 mM HEPES, pH = 7.7, 100 mM KCl, 1 mM MgCl2, 0.1 mM CaCl2, 5 mM EGTA, 2% sucrose). Washed beads were then incubated under constant rotation with the 15 mL of Halo-human γTuNA for 16 hr overnight at 4 °C. The beads were collected next day via a magnetic stand and washed with 3 CV of CSF-XB and resuspended to 2 mL with CSF-XB, 2% sucrose. Beads were either used fresh or stored for up to a week at 4 °C.

The next day, frozen extract (4 mL) was thawed in a room-temperature water bath. A total of 500 µL of clean, uncoupled Halo Magne beads were washed with MilliQ water and 3 CV of CSF-XB, 2% sucrose for pre-clearing of extract. Thawed extract was then incubated with the 500 µL of uncoupled Halo beads for 30 min at 4 °C with rotation to remove non-specific binders. The pre-clearing beads were removed with a magnet, and the now 'pre-cleared' extract was used to resuspend the 2 mL equivalent of coupled Halo-human γTuNA beads. Halo-human γTuNA beads were incubated with the extract for 2 hr, 4 °C, while rotating. The beads were then removed from the extract, washed with 4.5 CV of CSF-XB, and if required, incubated with 40 µM NHS-PEG4-Biotin (cat. #A39259, ThermoFisher) for 1 hr on ice and then resuspended with 2 mL of "3C" Elution buffer (600 µg PreScission "3C" protease diluted in CSF-XB, 2% sucrose). Proteins were eluted from the beads overnight at 4 °C, 15–18 hr, with rotation.

The elution containing cleaved γTuNA, γTuRC, and other factors, was concentrated to ~300 µL in a 100 kDa MWCO Amicon 4 mL spin concentrator. This was then spun through a 10–50% w/w sucrose gradient (in CSFxB buffer) using a TLS55 rotor in a Beckman Coulter Optima MAX-XP ultracentrifuge at 200,000 g, 2 °C for 3 hr. The gradient was manually fractionated such that the first fraction was the same size as the input (~300 µL), and each subsequent fraction was 140 µL in size. Samples of each fraction were run on a 4–12% Bis-TRIS SDS-PAGE gel for 10 min, 140 V. The SNAP i.d. 2.0 rapid Western blotting system (cat. #SNAP2MM, EMD-Millipore, Burlington, MA) was used to determine the peak γTuRC fraction (usually Fraction 7; γ-tubulin, GTU-88 mouse antibody, cat. # T6557, Sigma, St. Louis, MO). A 5 µL sample of the peak fraction was then added to glow discharged copper CF400-CU EM grids (cat. # 71150, Electron Microscopy Sciences, Hatfield, PA) for 1 min, and stained with 0.75% Uranyl Formate (UF) for 40 s. The grid was then imaged on a Philips CM100 transmission electron microscope at ×64,000 magnification, 80 kV, to verify the presence of intact γTuRCs, as well as to assess purity. A representative image is shown in *Figure 2*. The peak γTuRC fraction was stored on ice and used within 24 h or snap-frozen and stored at –80 °C. Our prep yielded an average peak concentration between 150–200 nM γTuRC (per 140 µL fraction). This was determined via Western blots probing for GCP4 in our peak fraction, which was compared against a known standard of recombinant GCP4. As there are two copies of GCP4 in each γTuRC, we divided our measured [GCP4] by two to get peak [γTuRC]. Via a fused Halo-AU1 reporter version of human γTuNA, we measured 50 nM γTuNA dimer in the peak γTuRC fraction, suggesting 67% of purified γTuRCs lost their γTuNA dimer

bait at the end of the sucrose gradient step (*Figure 2—figure supplement 2*). This loss of γTuNA bait has been previously observed (*Choi et al., 2010*; *Muroyama et al., 2016*).

## Negative stain EM data processing

Data processing of negative-stain EM data was done using Relion 3.1 (*Zivanov et al., 2018*). We manually picked 4866 total particles from 59 micrographs, followed by particle extraction and 5 rounds of 2D class averaging for particle sorting prior to 3D reconstructions. 4,593 particles were used for *ab initio*, non-templated reconstructions. Particles were further sorted with 3D class averaging, and 2,692 selected particles were used for structure refinement. Final refinement iterations were done using a 100 Å low-pass filtered mask with a contour cutoff of 0.007. This gave a final mask-sharpened map at 28 Å resolution as determined by the gold-standard FSC cutoff of 0.143 (*Figure 2—figure supplement 3*). CryoEM movie data was aligned and CTF corrected using CryoSparc 3.2 (*Punjani et al., 2017*), followed by subsequent manual picking of particles on CTF-corrected micrographs. A total of 800 particles were manually picked, extracted, and used to calculate 2D class averages.

## Pull-downs of γTuRC from *Xenopus* egg extract via Halo-γTuNA constructs

To compare γTuRC binding across mutant versions of γTuNA, we performed pulldowns from *Xenopus* egg extract with Halo-Magne beads (cat # G7281, Promega) coupled to N-terminally Halo-tagged versions of either human wildtype, *Xenopus* wildtype, or *Xenopus* dimer mutant γTuNA (as in *Figure 3*). Mock beads (blocked with bovine serum albumin) were used to assess background levels of non-specific precipitants. For each condition, we diluted 2 mg total of bait protein in 1 mL of Coupling Buffer (see section 6.5). Next, we resuspended 170 μL worth of Halo-Magne beads (washed 3 x with Coupling Buffer) with each protein mix. These were incubated under rotation at 4 °C for 1 h. Beads were then collected via a magnetic stand, washed three times with Coupling Buffer, and then resuspended to 1 mL final volume with CSFxB (2% sucrose).

Next, 2 mL total of frozen *Xenopus* egg extract were thawed in a room-temperature water bath. Beads for each condition were then collected via a magnetic stand and resuspended with 200 μL of extract. To each, we then added 800 μL of CSFxB (2% sucrose), and incubated under rotation at 4 °C for 2 hr. Beads were then washed twice with 1 mL of CSFxB (2% sucrose), prior to resuspension in 120 μL of 1 x SDS-PAGE sample buffer (6 mM DTT). Beads were then boiled at 95 °C for 5 min. After magnetic removal of beads, the elutions were spun at 17,000 g for 1 min to remove aggregates or beads.

We ran 40 μL of each elution per lane in 4–12% Bis-Tris SDS-PAGE gels at 140 V for 1 hr. Proteins were then transferred to nitrocellulose membranes and probed for γTuRC components via Western blot: GCP5 (1:250 dilution of mouse anti-GCP5 antibody (E-1), # sc-365837, Santa Cruz Biotechnology, Santa Cruz, CA; RRID:AB_10847352) and γ-tubulin (1:1000 dilution of GTU-88 mouse antibody, cat. # T6557, Sigma; RRID:AB_2863751). To confirm equal coupling of bait to beads, we also probed for the Strep-tag on γTuNA (1:1000 dilution of *Strep-tag* mouse monoclonal antibody, cat. # 34850, Qiagen, RRID:AB_2810987). Band intensities were measured in ImageJ and normalized in Prism 7 by the wild-type γTuNA band (positive control). Independent experiments, after normalization, were averaged together into the charts seen in *Figure 3* (N=3 for *Figure 3F*, N=2 for *Figure 3G*).

## Single molecule TIRF imaging of γTuRC MT nucleation in vitro
### Preparation of functionalized coverslips and imaging chambers

We utilized a previously published method to generate functionalized glass coverslips for single molecule TIRF imaging (*Thawani et al., 2020*; *Consolati et al., 2020*). Briefly, after sonication with 3 M NaOH, coverslips were sonicated with Piranha solution (2:3 ratio of 30% w/w $H_2O_2$ to sulfuric acid) to remove all organic residues. After washes with MilliQ water, we dried and then treated the coverslips with 3-glycidyloxypropyl trimethoxysilane (GOPTS, cat. # 440167, Sigma) at 75 °C for 30 min. Unreacted GOPTS was removed with two sequential acetone washes, and the coverslips were then dried with nitrogen gas. Between sandwiched coverslips, we melted a powder mix of 9 parts HO-PEG-NH₂ and 1 part biotin-CONH-PEG-NH₂ by weight (cat. #103000–20 and #133000-25-20, Rapp Polymere, Tübingen, Germany) at 75 °C. We pressed out air bubbles and repeated cycles of 75 °C incubation and pressing until sandwiches were clear. After an overnight incubation at 75 °C, the sandwiches were

separated and washed in MilliQ water. These were then spun dry in air and stored at 4 °C for up to 1 month.

We made imaging chambers by first making a channel on a glass slide with double-sided tape. To this channel, we added 2 mg/mL PLL-g-PEG (SuSOS AG, Dübendorf, Switzerland) in MilliQ water. After a 20 min incubation, the channel was rinsed thoroughly with MilliQ water and dried with nitrogen gas. Using a diamond pen, functionalized coverslips were cut into quarters. Each quarter piece was then added to the double sided tape on the slide with the functionalized surface facing down. Chambers were made fresh on the day of each experiment.

## Attaching biotinylated, Halo-prepped γTuRC to functionalized chambers

As previously described (*Thawani et al., 2020*), imaging chambers were blocked first with 50 µL of 5% w/v Pluronic F-127, then 100 µL of assay buffer (BRB80 with 30 mM KCl, 1 mM GTP, 0.075% w/v methylcellulose 4000 cp, 1% w/v D-glucose, 0.02% w/v Brij-35, and 5 mM BME). Next, we added 100 µL of casein buffer (0.05 mg/ml κ-casein in assay buffer). Here, we modified our protocol from our previous study by adding 20 µL of 0.05 mg/mL NeutrAvidin (A2666, ThermoFisher), not 50 µL of 0.5 mg/mL. We also decreased the cold block incubation at this step from 3 min to 1.5 min. We then washed the channel with 100 µL of BRB80. Similarly, we diluted peak fraction Halo-prepped, biotin- γTuRC between 1/100 to 1/300 in BRB80 depending on the prep yield, not 1/5 as previously published. This was due to the increased yield of Halo-purified γTuRC as compared to our previous antibody-based method. We added 20 µL of this γTuRC dilution and incubated for 7 min at room temperature. Finally, the chamber was washed with 20 µL of cold BRB80. All buffers referenced here, except Pluronic F-127 (room-temperature), were used at 2 °C.

## Microtubule nucleation assay with purified γTuRC

Concurrently, we prepared our nucleation mix by mixing 15 µM total unlabeled bovine tubulin (PurSolutions, Nashville, TN) with 5% Alexa 568-tubulin, 1 mg/mL BSA (cat. # A7906, Sigma) in assay buffer on ice. To this mix, we also added either Tris control buffer or Strep-His-3C-*Xenopus* γTuNA constructs (to 3.3 µM final concentration). This was then centrifuged in a TLA100 rotor (Beckman Coulter) for 12 min at 80,000 rpm to remove aggregates. Finally, we added 0.68 mg/ml glucose oxidase (cat. # SE22778, SERVA GmbH, Heidelberg, Germany), 0.16 mg/ml catalase (cat. # SRE0041, Sigma). This nucleation mix was then added to the chamber with attached γTuRC and imaged immediately. For assays using stathmin/op18, we adjusted our nucleation mix so that it contained either stathmin control buffer (50 mM Tris, pH = 7, 300 mM NaCl, 400 mM imidazole) or 2.5 µM or 4 µM final concentration of 6xHis-SNAP-*Xenopus* stathmin (isoform 1 A).

We used the same imaging set-up as our previous work (*Thawani et al., 2020*), notably a Nikon Ti-E inverted stand (RRID:SCR_021242) with an Apo TIRF 100 x oil objective (NA = 1.49) and an Andor iXon DU-897 EM-CCD camera with EM gain set to 300. We again used an objective heater collar (model 150819–13, Bioptechs) to maintain 33.5 °C for our experiments. However, for this study we captured time-lapse movies of the tubulin 561 nm channel at 1 frame every 2 s for 5 min. All movies start within 1 min of the addition of ice-cold nucleation mix to the imaging chamber. Biological replicates were done with independent Halo-γTuRC preps: wildtype γTuNA reactions (n=8), buffer control (n=6), γTuNA-F75A (n=5), and γTuNA-L77A (n=3).

## Analysis of single molecule TIRF MT nucleation assays
### Analysis of total MT mass

For total MT mass measurements, we wrote a FIJI/ImageJ macro to measure the total 561 nm signal intensity for each frame in our time-lapses. To do this, the macro first filtered each frame using the Otsu method in the 'Adjust Threshold' function to remove most background signal. Next, it used the 'Measure' function and recorded the mean intensity for each frame. In MATLAB, we then subtracted the mean intensity of the first frame (as background) from all frames in our time-series and normalized for each condition by the buffer control. We then plotted MT signal (MT mass) over time, as shown in *Figure 5B*. For the assays in *Figure 6C–D*, we used the same method, except we plotted the total MT mass generated at 300 s, normalized by the buffer condition.

## Analysis of MT number over time, nucleation rate, growth speed, and mean MT length

For each time-series, we analyzed an area 40 μm x 40 μm (252x252 pixels$^2$) for the first 150 s of each reaction. We first corrected for minor translational drift in our movies by using the StackReg plugin for ImageJ (RRID:SCR_003070; *Thévenaz et al., 1998*). Next, we wrote two FIJI/ImageJ macros to semi-automate our data analysis. The first macro generated kymographs (space-time plots) for each individual MT in the time-lapse, although each MT was manually selected. With the second macro, we manually extracted relevant parameters from these kymographs.

First, if the MT was spontaneously nucleated, the resulting kymograph would display bi-directional growth over time (appearing like a scalene triangle). If the MT was nucleated by a γTuRC, then one end did not grow over time resulting in kymographs with only a single growing edge (right angle triangle). Using the macro, we recorded whether each MT was spontaneously or γTuRC nucleated. If the MT was γTuRC-nucleated, we proceeded with our measurements. We next manually recorded the nucleation point (or origin) for each MT. We then drew a line along the growing edge and extracted its slope to generate the growth speed for that MT. We also measured the MT's maximum length. These measurements were then imported into MATLAB (R2019a) and averaged across all reactions for each condition. The mean and standard error (SEM) for the number of MTs nucleated over time were plotted using MATLAB, as shown in *Figure 5C*. To determine the γTuRC nucleation rate, we fit the MT nucleation curves for each condition to *Equation 1*,

$$N\left(t\right) = N_{max} * \left[1 - e^{\frac{-kt}{N_{max}}}\right] \tag{1}$$

where $N(t)$=the number of MTs nucleated at that time point, $N_{max}$ = the maximum number of MTs nucleated after 150 s, and $k$=the γTuRC nucleation rate. We used the nonlinear least squares fitting algorithm from MATLAB's "lsqcurvefit" function to determine both $N_{max}$ and $k$. This nucleation rate ($k$) was then averaged for all reactions in each condition, including calculating the standard error of $k$. Both the mean and SEM for the γTuRC nucleation rate are shown in *Figure 5D*. We also plotted the linear slope of the curves at saturation (from 25 to 150s) in *Figure 5—figure supplement 1*. The distributions of our growth speed and mean maximum MT length measurements are shown in violin plots in *Figure 5E–F*. For *Figure 5C* through 5 F, two-sample, unpaired t-tests were used to determine if the means for each condition were significantly different from the buffer condition. Differences with p-values less than 0.05 were considered significant.

## Simulations of γTuNA's effect on γTuRC MT nucleation

To simulate the effect of γTuNA in our single molecule TIRF assay, we wrote a deterministic simulation in MATLAB based on the measured nucleation rates for each condition. For simplicity, we assume no spontaneous MT nucleation and no MT catastrophes. This system was modeled by *Equation 2*, as follows:

$$N\left(t\right) = N_{\left(t-1\right)} + k * \left(1 - \frac{N_{\left(t-1\right)}}{N_{max}}\right) \tag{2}$$

where $N(t)$=the current number of nucleated MTs (or active γTuRCs), $k$=the nucleation rate (MTs nucleated per second), and $N_{max}$ is the total number of γTuRCs activatable by that condition. At $N(t=0)$, the number of active γTuRCs or nucleated MTs is zero. At each time step (1 s), new MTs are added to the system according to the nucleation rate measured experimentally ($k$), which decreases until reaching saturation. These new MTs were then randomly placed on a simulated 40x40 pixel$^2$ plane with a random initial orientation based on one of eight discrete conditions. At each new time step, the length of previous MTs is incremented by a constant growth speed (the mean speed from all conditions in *Figure 5E*, as μm/s). This process of nucleation and growth is repeated until the end of the simulation, generating simulated movies of this process. For the L77A mutant, we also generated a second two-step simulation where, after 150 s, $k_{L77A}$ is arbitrarily redefined as the wildtype $k$ ($k_{WT}$), and $N_{max}$ in *Equation 2* is redefined as (wildtype $N_{max}$ – L77A $N_{(150s)}$), where L77A $N_{(150s)}$=the number of MTs already present at 150 s. Plots and simulated frames are shown in *Figure 5—figure supplement 3*.

## Mass spectrometry identification of unique *Xenopus laevis* γTuNA binding factors

### Sample preparation

We performed pulldowns from *Xenopus* egg extract with Halo-Magne beads (Promega cat # G7281, Madison, Wisconsin, USA) coupled to either human wildtype, *Xenopus* wildtype, or *Xenopus* F75A mutant versions of Halo-γTuNA (*Figure 2—figure supplement 1*). Uncoupled beads were used to assess background levels of non-specific precipitants. After washing, bound proteins were eluted with Glutathione-S-transferase (GST) tagged PreScission (HRV 3 C) protease. Elutions were then subjected to a reverse GST step to remove most GST-PreScission. Ten percent of each elution sample was run on an SDS-PAGE gel and stained with Coomassie to confirm low levels of non-specific protein binders (bead control, *Figure 2—figure supplement 1B*). We also probed these elutions for the presence of γTuRC components, GCP5 and γ-tubulin, confirming they were only present when the bait was a wildtype version of γTuNA (*Figure 2—figure supplement 1C*). This pulldown was performed two times independently to generate a set of six samples (two replicates for each γTuNA condition) that were then submitted to the ThermoFisher Center for Multiplexed Proteomics (TCMP) for multiplexed quantitative mass spectrometry (Harvard Medical School, Boston, MA).

### Quantitative mass spectrometry (MS; Quant-IP)

At TCMP, the concentrations of our six samples were measured via a Pierce micro-BCA assay. Samples were then reduced with DTT and alkylated with iodoacetimide. This was followed by a protein precipitation step using methanol/chloroform. The resulting pellets were resuspended in 200 mM EPPS, pH 8.0. Samples were then digested sequentially using LysC (1:50) and Trypsin (1:100), determined by the protease to protein ratio. The digested peptides from each condition were then separately labeled with one of six tandem mass tags (TMT) for multiplexing (TMT-126, TMT-127a, TMT-127b, TMT-128a, TMT-128b, TMT-129a). All samples were then combined and run through basic pH reverse phase (bRP) sample fractionation utilizing an 8-to-28% linear gradient of acetonitrile (ACN; in 50 mM ammonium bicarbonate buffer, pH 8.0). These fractionated, TMT-tagged peptides were then analyzed via three sequential mass spectrometry scans (LC-MS3): a precursor ion Orbitrap scan (MS1), followed by an ion trap peptide sequencing scan (MS2), and a final Orbitrap scan to quantify the reporter ions (MS3).

### Database search parameters

All MS2 spectra were analyzed using the Sequest program (Thermo Fisher Scientific, San Jose, CA, USA). Sequest was used with the following search parameters: peptide mass tolerance = 20 ppm, fragment ion tolerance = 1, Max Internal Cleavage Sites = 2, and Max differential/Sites = 4. Oxidation of methionine was specified in Sequest as a variable modification. MS2 spectra were searched using the SEQUEST algorithm with a Uniprot composite database derived from the *Xenopus* proteome containing its reversed complement and known contaminants. Peptide spectral matches were filtered to a 1% false discovery rate (FDR) using the target-decoy strategy combined with linear discriminant analysis. Identified proteins were filtered to a<1% FDR. Proteins were quantified only from peptides with a summed SN threshold of >100 and MS2 isolation specificity of 0.5. From this, 21,902 unique peptides were detected, resulting in 3,214 total proteins. After filtering, this resulted in 2,842 unique, quantified proteins across our six γTuNA samples. The top 12 proteins (with at least 5 unique peptides) specifically enriched in the wildtype γTuNA samples are presented in *Figure 2—figure supplement 1*.

### Statistical analysis

Two-sample, unpaired Student's t-tests were used with significance declared when p-values were less than p=0.05. No a priori sample size or power analysis calculations were performed. For extract or γTuRC in vitro experiments, the number of datasets (or 'N') refers to biological replicates, performed with either independent *Xenopus* egg extracts or independent Halo-γTuRC preps from different *Xenopus* egg extracts. For all main figure data, standard error of the mean (SEM) is used to indicate uncertainty in our measurement of each condition's mean, except for *Figure 5E and F*, where the complete distributions of growth speed and MT length measurements are instead shown for clarity.

## IACUC approved use of laboratory animals: *Xenopus laevis* frogs

Experimental use of *Xenopus laevis* frogs was done in strict accordance with our approved Institutional Animal Care and Use Committee (IACUC) protocol # 1941–16 (Princeton University).

## Acknowledgements

The authors thank all members of the Petry lab, past and present. In particular, we would like to thank Dr. Raymundo Alfaro-Aco. We also thank Prof. Fred Hughson (Princeton University), Prof. Paul Conduit (Institut Jacques Monod), and Bernardo Gouveia (Princeton University) for their insightful comments on our manuscript. We also thank Dr. Jodi Kraus for sharing reagents. We especially thank both the Princeton Mass Spectrometry core facility and the ThermoFisher Center for Multiplexed Proteomics (TCMP) at Harvard Medical School. We also thank the Imaging and Analysis Center (IAC) at Princeton University, which is partially supported by the Princeton Center for Complex Materials (PCCM), a National Science Foundation (NSF) Materials Research Science and Engineering Center (MRSEC; DMR-2011750), and in particular, Dr. John Schreiber. We also thank Dr. Ron Vale (UCSF) for the plasmid containing GCN4 (Addgene plasmid# 74608).

## Additional information

### Funding

| Funder | Grant reference number | Author |
| --- | --- | --- |
| Howard Hughes Medical Institute | Gilliam Graduate Student Fellowship | Michael J Rale |
| National Science Foundation | Graduate Research Fellowship | Michael J Rale |
| National Institutes of Health | New Innovator Award | Sabine Petry |
| Pew Charitable Trusts | Pew Scholars Program in the Biomedical Sciences | Sabine Petry |
| David and Lucile Packard Foundation | 2014-40376 | Sabine Petry |
| National Institutes of Health | 1DP2GM123493 | Sabine Petry |

The funders had no role in study design, data collection and interpretation, or the decision to submit the work for publication.

### Author contributions

Michael J Rale, Conceptualization, Software, Formal analysis, Funding acquisition, Investigation, Visualization, Methodology, Writing - original draft, Writing - review and editing; Brianna Romer, Brian P Mahon, Sophie M Travis, Investigation, Visualization, Writing - review and editing; Sabine Petry, Conceptualization, Resources, Formal analysis, Supervision, Funding acquisition, Visualization, Methodology, Writing - original draft, Project administration, Writing - review and editing

### Author ORCIDs

Michael J Rale ⓘ http://orcid.org/0000-0003-1426-6611
Brianna Romer ⓘ http://orcid.org/0000-0003-1772-4243
Brian P Mahon ⓘ http://orcid.org/0000-0002-5571-8058
Sophie M Travis ⓘ http://orcid.org/0000-0002-1728-1705
Sabine Petry ⓘ http://orcid.org/0000-0002-8537-9763

### Ethics

Experimental use of Xenopus laevis frogs was done in strict accordance with our approved Institutional Animal Care and Use Committee (IACUC) protocol # 1941-06 (Princeton University).

Decision letter and Author response
Decision letter https://doi.org/10.7554/eLife.80053.sa1
Author response https://doi.org/10.7554/eLife.80053.sa2

## Additional files

### Supplementary files

• Supplementary file 1. Primers used to generate γTuNA constructs used in this study; pertains to *Table 1*.

• MDAR checklist

• Source code 1. MATLAB source code for numerical simulation of MT nucleation from purified γTuRCs in the presence of γTuNA constructs (used to generate *Figure 5—figure supplement 3* and Video 3).

• Source code 2. MATLAB source code for graphical simulation of MT nucleation from purified γTuRCs in the presence of γTuNA constructs (uses *Source code 1* as input; used to generate *Figure 5—figure supplement 3* and Video 3).

### Data availability

Raw and processed microscopy data, related analysis scripts (ImageJ and MATLAB), raw size-exclusion chromatography files, and mass spectrometry data have been deposited in a freely accessible dataset on Dryad (Dataset DOI: https://doi.org/10.5061/dryad.gb5mkkwt3). Figure source data and MATLAB code are also included in this study as supplemental or source data files. Plasmids generated in this study are available upon request from the corresponding author.

The following dataset was generated:

| Author(s) | Year | Dataset title | Dataset URL | Database and Identifier |
|---|---|---|---|---|
| Rale MJ, Romer B, Mahon B, Travis S, Petry S | 2023 | Data for: The conserved centrosomin motif, γTuNA, forms a dimer that directly activates microtubule nucleation by the γ-tubulin ring complex | https://dx.doi.org/10.5061/dryad.gb5mkkwt3 | Dryad Digital Repository, 10.5061/dryad.gb5mkkwt3 |

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
