## [Editor Report]

This fundamental Research Advance is of interest to cell biologists studying the mechanisms and control of microtubule nucleation. Rale et al. convincingly establish the regulatory role of the γ-TuNA motif in microtubule nucleation and settle prior conflicting results in the literature. They show that γ-TuNA binds to and activates γ-TuRC-based microtubule nucleation both in *Xenopus* extracts and in vitro.

---

## [Decision Letter]

**Decision letter after peer review:**

Thank you for submitting your Research Advance article "The conserved centrosomal motif, γTuNA, forms a dimer that directly activates microtubule nucleation by the γ-tubulin ring complex (γTuRC)" for consideration by *eLife*. Your article has been reviewed by 3 peer reviewers, and the evaluation has been overseen by Suzanne Pfeffer as the Senior Editor. The reviewers have opted to remain anonymous.

The reviewers are in favor of publication of your story after you make textual changes to address the comments they have listed in their reviews.

*Reviewer #1 (Recommendations for the authors):*

1) Is it an issue that a third of purified γ-TuRCs still have γ-TuNA attached? this would suggest that a third of γ-TuRCs in any mock experiment would be active, but that doesn't seem to be reflected in the data…? While they have used human γ-TuNA to purify *Xenopus* γ-TuRCs, I suspect that human γ-TuNA would also activate *Xenopus* γ-TuRCs (or maybe that is a bad assumption). Please discuss.

2) In the title the authors use "Centrosomal motif, γ-TuNA" but I'm not sure it is fair to refer to it as a centrosomal motif, given that it is found in proteins that recruit γ-TuRCs to various MTOCs. CM1 refers to "Centrosomin motif 1", rather than centrosomal motif.

3) In the abstract, the authors say they "build on" their previous study, but I would argue that they are not building on it, they are correcting a major conclusion from it. Also, in the abstract they say that they "illuminate how γ-TuRC is controlled in space and time in order to build specific cytoskeletal structures" (and this statement is essentially repeated in the first line of the Discussion), but the authors have not studied these aspects at all. This, therefore, needs to be removed/toned down.

4) It would be nice for the authors to improve the diagrams in Figure 1a/b, making it clearer where the sequence of γ-TuNA is located within the full length protein. They should include amino acid numbers in Figure 1b and also for all their constructs in Figure 1 supp 1 (not just 34aa fragment, etc). Is F75 really F75 in *Xenopus*? Which *Xenopus* isoform is being used?

5) Regarding F75A mutants: From the images in Figure 2B, and without seeing any quantification, it seems that there is some aster formation above the mock control when using the F75A fragments, suggesting that either the F75A mutant is capable of binding a low level of γ-TuRCs or that the F75A fragment binds something else that may promote a low level of MT nucleation. It seems unlikely that it F75A fragments are binding a low level of γ-TuRCs, given the data in Figure 3d/e. Based on the mass spec data, could γ-TuNA be allowing NME7 to bind? This could also explain why microtubule nucleation is higher with F75A than with other mutants in Figure 4B. I'm not expecting the authors to perform extra experiments for this, but a discussion would be nice.

6) The authors discuss the idea that the effect of γ-TuNA could be further enhanced by γ-TuRCs being recruited to sites of high local tubulin concentration. Again, this would be easy for them to test, by varying tubulin concentration in their assays and measuring the fold change in γ-TuRC-mediated nucleation in mock vs γ-TuNA addition.

*Reviewer #2 (Recommendations for the authors):*

Overall, I think this is a well-controlled study. However, the discussion that emphasizes the importance of γ-TuNA-dimerization, as a pre-requisite for the binding and activation of γ-TuRCs, has some flaws that should be addressed:

1) The human double mutant I67A/L70A has a strong negative effect on γ-TuNA dimerization (Figure 3B), but still enables the binding to γ-TuRCs (Figure 3E).

2) The mutant F75A is perfectly able to dimerize, but fails to bind γ-TuRCs and fails to stimulate microtubule nucleation.

These discrepancies are neglected in the results and Discussion sections. They suggest that failure of dimerization may be overcome by certain mutations such as I67A/L70A, and they suggest that the γ-TuNA external protein surface (not involved in the coiled-coil) is also essential for the binding to the γ-TuRC.

*Reviewer #3 (Recommendations for the authors):*

The present study focuses on the effect of a specific structural motif γ-TuNA in microtubule formation. The study is an important advance to widen the discussion around the spatiotemporal regulation of microtubule nucleation through γ-TuNA-motif-containing proteins. Although the activating function of γ-TuNA was reported earlier, there were contrasting results in the literature that have been resolved by this study. Another advance is a scaled-up method to purify γTuRC using Halo- γ-TuNA and validation of the purification through in vitro nucleation experiments and electron microscopy. While the study does not provide new mechanistic insights to understand how γ-TuNA affects γTuRC and whether the observed effects on nucleation are a general feature of γ-TuNA motifs in the context of different proteins such as CDK5RAP2, TPX2, and CEP215, the work here is important for the field as suitable for publication as an *eLife* advances report. The following needs to be addressed in the text prior to publication.

1. The authors make a strong claim that the dimeric state of γ-TuNA is necessary for binding γ-TuRC and activating microtubule nucleation. Although the authors show that the double and triple mutants show loss of dimerization and binding to γ-TuRC, there are multiple exceptions that cannot be explained and should be discussed explicitly:

a. L70A mutant does not fully dimerize (comparable to L77A) but can still bind γ-TuRC (Figure 3).

b. GCN4-tagged mutants that form a dimer cannot bind γ-TuRC (Fig-3-Suppl-1).

c. The most important of all exceptions is the L77A mutant that is neither a complete dimer, nor does it bind γ-TuRC (Figure 3) but is still able to activate microtubule nucleation at longer time points (Figure 5).

d. Related to (c), the authors state that "Interestingly, we found that the intermediate dimer mutants (I67A, L70A, or L77A) had correspondingly intermediate levels of γTuRC" (line 149), but the dimerization capability of L70 and L77 is similar and yet very different in γ-TuRC binding.

e. L70A is missing in Figure-5 and having that information will be helpful in making correlative conclusions (Not Essential).

Overall, I am convinced that the residues in the coiled-coil are important for γ-TuRC binding but not fully convinced about the dimerization claim without an experiment where forced dimerization restores activity in a mutant. The writing should be edited for greater precision.

2. The study discusses that the effect of γ-TuNA on γTuRC is transient and might depend on local tubulin concentrations. Related to this point, the authors explicitly show in their model that γ-TuNA counteracts the effects of stathmin by increasing the efficiency of nucleation at lower tubulin concentrations (Fig6). This hypothesis can be easily tested in their in vitro assay by examining if γ-TuNA decreases the critical conc. of tubulin required for γ-TuRC-based microtubule nucleation. This would provide a mechanism for γ-TuNA's effects and strengthen the paper. If this cannot be done, then I suggest removing this specific mechanistic detail from the model figure.

3. The γ-TuRC purified using γ-TuNA-coated beads followed by sucrose gradient step still retains ~30% wild type γ-TuNA (methods Line 486-488). This should be explicitly stated in the main text as it is very relevant to experiments where mutant γ-TuNA is added, and the effects observed are on top of the wild-type γ-TuNA from the purification.

4. In Figure 1F, the authors deplete γTuRC from egg extracts and conduct microtubule nucleation experiments to show the dependence of γ-TuNA on the γTuRC based templates to build microtubules. Although this experiment was effective in proving the point, the authors should explain their decision to choose γ-TuNA coupled beads to deplete γTuRC from the extracts. Depleting γTuRC using any other mechanism, for example, anti-γ tubulin antibody coupled beads should also show similar effects.

5. The authors write in the discussion that γ-TuNA fails to activate antibody purified γTuRC. This is extremely puzzling. Does a comparison of the two preps γTuRC (antibody-based purification and γTuNA-based purification) by mass spec provide any hints to additional factors? If there was an additional activator in the γ-TuNA-based purification, then the effects observed in this paper are not entirely due to the γ-TuNA.

6. The authors utilize purified γ-TuNA motif from CDK5RAP2 in this study. The authors indicate that γ-TuNA motif has no role in γTuRC assembly through fractionation (pg, 4, line 90-92 and Figure 1-Suppl-2), drawing a clear distinction between γ-TuNA and other nucleation activators. However, in the context of full-length CDK5RAP2, the γ-TuNA motif may very well have a role in γTuRC assembly. So, this conclusion seems like a stretch. I recommend changing the text. Related to this, ideally "γ-TuNA motif or domain" would be a better phrase choice throughout the paper instead of γ-TuNA.

7. In Fig-2-supp-1-B, please label the bands on the gel that are of interest.

[Editors’ note: further revisions were suggested prior to acceptance, as described below.]

Thank you for resubmitting your work entitled "The conserved centrosomin motif, γTuNA, forms a dimer that directly activates microtubule nucleation by the γ-tubulin ring complex (γTuRC)" for further consideration by *eLife*. Your revised article has been evaluated by Suzanne Pfeffer (Senior Editor) and a Reviewing Editor.

The manuscript has been improved but there are some remaining textual issues that need to be addressed, as outlined below:

The reviewers feel strongly that the replies to their comments be included in the actual manuscript and they request that the Abstract be modified to precisely reflect the conclusions of the work. When resubmitting, please include a track-changes version to expedite the re-review process.

*Reviewer #1 (Recommendations for the authors):*

I think the authors have made a good effort to address the reviewer's concerns and the paper is nearly ready for publication. Nevertheless, I still have a few concerns that I hope the authors can take into account.

Reviewer point 3:

In the abstract, the authors say they "build on" their previous study, but I would argue that they are not building on it, they are correcting a major conclusion from it. Also, in the abstract they say that they "illuminate how γ-TuRC is controlled in space and time in order to build specific cytoskeletal structures" (and this statement is essentially repeated in the first line of the Discussion), but the authors have not studied these aspects at all. This, therefore, needs to be removed/toned down.

Author Response:

These are excellent points raised by the reviewer. We agree that in essence we are correcting a conclusion from our previous study concerning the role of γTuNA on γTuRC activity and consequently, we edited the abstract. As to the spatial-temporal regulation of the level of γTuRC activity, we also agree that most of our data concerning the interaction of γTuNA with γTuRC, the effect on γTuRC activity, and the antagonistic role of stathmin are largely addressing the spatial regulation of microtubule nucleation (MTOC vs cytoplasm). However, previous work has shown that the number of CDK5RAP2 molecules at the centrosome (the most potent MTOC) increases during the transition from interphase to mitosis which results in increased MT nucleation activity via increased γTuRC recruitment (see Piehl et al., PNAS, 2004; Lawo et al., Nat Cell Biol, 2012; and Menella et al., Nat Cell Biol, 2012). Thus, the temporal control of the number of γTuRC binding sites (in interphase vs mitosis) is a method of temporally regulating the level of microtubule nucleation, while simultaneously spatially restricting it. We see our findings here as supportive of the hypothesis that controlling the cellular location and number of γTuNAs over time is a means of spatially and temporally regulating γTuRC microtubule nucleation. On top of that, and critically, we directly show that we can immediately activate γTuRC activity via the addition of dimerized γTuNA, thus showing that the interaction of γTuRC and γTuNA at the centrosome are essentially an intertwined form of spatial and temporal control. To address these points, we edited the discussion in a section on spatio-temporal control (page 18).

New Reviewer Response:

I appreciate the authors additional section in the discussion about the work that others have carried out on the spatiotemporal control of γ-TuRC activation. However, it seems the authors have misunderstood my original point about not showing data on the spatiotemporal regulation of γ-TuRC activation – I did not mean that they had only addressed spatial control; I meant that they had not addressed either spatial or temporal control. This is because their data shows, very nicely, that γ-TuNA binding to γ-TuRCs activates γ-TuRCs but it does not show how γ-TuNA binding to γ-TuRCs is regulated in space and time. The authors appear to propose that spatial control is achieved by localising CDK5RAP2 to centrosomes, but CDK5RAP2 is also present in the cytoplasm and so, in theory, has the potential to bind (and activate) γ-TuRCs in the cytoplasm. As the authors discuss, this binding is likely auto-inhibited by other cis-regulatory elements, but this idea comes from other studies, not this one. And they certainly have not linked this to building "specific cytoskeletal structures". I think the authors should focus their summary statement on what they have actually shown – something more like "In sum, our improved assays finally prove that γ-TuNA binding strongly activates γ-TuRCs, explaining previously observed effects of γ-TuNA expression in cells".

Reviewer point 6:

The authors discuss the idea that the effect of γ-TuNA could be further enhanced by γ-TuRCs being recruited to sites of high local tubulin concentration. Again, this would be easy for them to test, by varying tubulin concentration in their assays and measuring the fold change in γ-TuRCmediated nucleation in mock vs γ-TuNA addition.

Author Response:

We thank the reviewer for their comment. We agree that varying the tubulin concentration is a worthwhile experiment, especially in light of our hypothesis that γTuNA enables γTuRC to nucleate MTs at lower tubulin concentrations than otherwise possible (as suggested by our in vitro experiments using stathmin in Figure 6). While we could increase the tubulin concentration above 15 µM tubulin in the in vitro assay, we do not report higher concentrations in this study as background spontaneous MT nucleation increases such that it can become difficult to distinguish γTuRC-nucleated MTs. Also, at 15 µM tubulin and 1.65 µM γTuNA dimer (3.3 µM monomer γTuNA) we find our assay saturates within 25 seconds, by which time we see a maximum of ~300 MTs per 40 µm by 40 µm area on the coverslip (Figure 5C). This is in agreement with the number of fluorescent γTuRCs we see for the equivalent surface area (mean: 377 MTs; first to third quartile range: 302 to 469 γTuRCs; see Author Response Figure 1). For this reason, we believe that practically all nucleation-competent γTuRCs attached to our coverslips are nucleating MTs within 25 seconds in the presence of 15 μm tubulin and 1.65 µM γTuNA dimer (3.3 µM monomer). Thus 15 µM is the highest we can increase the tubulin concentration in our assays. As a result, we decided to instead decrease our tubulin concentration in the assay, setting it at γTuRC's critical concentration. We then assayed for γTuNA's effect on γTuRC MT nucleation at this low tubulin concentration in vitro (now incorporated into Figure 6, as panel E). In our previous study (Thawani et al., 2020), we found that the critical concentration in the presence of γTuRC was 7 µM tubulin, where γTuRC has very low but still detectable nucleation ability (at least 1 MT nucleated per field during the course of a 5 min experiment). This was also observed by our colleagues in Consolati et al., Dev Cell, 2020 at 7.5 µM tubulin (defined by the authors as at least 1 MT nucleated per 164 µm^2^ field over 20 min). As shown in Figure 6E, we again validate this finding, observing that our purified γTuRC has very little activity at 7 µM tubulin with only 1 to 2 MTs nucleated per field of view over the course of 5 min (mock buffer condition). These MTs are also short and appear to be more dynamic than those generated by γTuRC at 15 µM tubulin. In Figure 6E, we show all MTs nucleated in our assay at 7 µM tubulin, regardless of lifetime, by presenting max intensity projections for our 5 min time series. Critically, in the presence of 1.65 µM γTuNA dimer (3.3 µM monomer), ~40 γTuRCs nucleate MTs per field despite being at this low tubulin concentration. This is approximately a 25-fold increase in the total number of MTs generated over the entire 5 min assay, as compared to the buffer condition (Figure 6E). Thus, we find further validation that γTuNA enables γTuRC to nucleate MTs even under low, constrained tubulin concentrations, an observation we first described in our experiments with stathmin (Figure 6A-D). We would also like to note that this new experiment is conceptually equivalent to our Figure 6C experiments using 15 µM tubulin and 4 µM stathmin. As 1 molecule of stathmin sequesters 2 tubulin subunits, 4 µM stathmin would leave only ~7 µM tubulin free in a 15 µM tubulin reaction (assuming 100% of the stathmin is functional). The fact that these two experiments agree not only confirms that γTuNA increases γTuRC's efficiency (i.e. lowers γTuRC's critical concentration), but also that stathmin indirectly regulates γTuRC activity by constraining the cytoplasmic tubulin pool.

New Reviewer Response:

I appreciate the extra experiments, but I'm not sure they address the initial point, which was whether "the effect of γ-TuNA could be further enhanced by γ-TuRCs being recruited to sites of high local tubulin concentration", and not whether γ-TuNA promotes microtubule nucleation at low tubulin concentrations (which it clearly does). The authors actually make the point that raising the tubulin concentration in their experiments makes it hard to distinguish the microtubules nucleated by γ-TuRCs from those nucleated spontaneously. From this, I would assume that the effect of γ-TuNA would not be further enhanced when γ-TuRCs are recruited to sites of high local tubulin concentration, because microtubules would simply spontaneously nucleate if the tubulin concentration were high enough. It seems more valid to report that γ-TuNA is required when tubulin concentrations are low.

*Reviewer #2 (Recommendations for the authors):*

As I stated already in my earlier review, I am in favor of this manuscript.

I am a bit disappointed, though, by the manner that this revision has been performed: the authors have provided detailed responses to my concerns in their rebuttal letter (importance of dimerization for gTuRC-binding; gTuNA mutants), but it seems that they haven't significantly addressed these points in the revised manuscript. Similarly, they ignored to discuss related concerns raised by reviewer 3. Our comments were meant to help improving the manuscript! Simply arguing in the rebuttal is not enough…

---

## [Author Response]

Reviewer #1 (Recommendations for the authors):1) Is it an issue that a third of purified γ-TuRCs still have γ-TuNA attached? this would suggest that a third of γ-TuRCs in any mock experiment would be active, but that doesn't seem to be reflected in the data…? While they have used human γ-TuNA to purify Xenopus γ-TuRCs, I suspect that human γ-TuNA would also activate *Xenopus* γ-TuRCs (or maybe that is a bad assumption). Please discuss.

We thank the reviewer for their insightful comment. From our quantifications, we do find that 50 nM human γTuNA dimer is present in the peak γTuRC fraction after the sucrose gradient step of our purification. Based on our measurements of GCP4’s concentration in this same fraction, we found that γTuRC’s concentration was on average 150 nM. Assuming only 1 γTuNA dimer binds per γTuRC (as suggested by the structure from Wieczorek et al., Cell, 2020), we estimate that at maximum one-third of the γTuRCs in our peak fraction have a γTuNA dimer attached.

Thus, the majority of γTuNA bait is lost by γTuRCs over the course of the purification. This observation was first described by our colleagues in Choi et al., 2010 and later by Muroyama et al., 2016. More recently, the Kapoor group (in Wieczorek et al., *Cell* 2020) similarly used an N-terminally tagged γTuNA as bait for their γTuRC. They, like Choi et al. and Muroyama et al., did not report how much of their γTuRC might be occupied by γTuNA (perhaps due to sub-nM yields of γTuRC). However, they did later demonstrate that enough γTuRCs are occupied by a dimer of γTuNA to generate a cryo-EM structure of the dimer in the complex (Wieczorek et al., *Cell Reports*, 2020). It is therefore likely that all past work involving use of CM1/γTuNA as a bait for γTuRC contained some small but persistent amount of γTuNA at the end of the purification, perhaps undetected until now as a result of low yields of γTuRC.

We agree with the reviewer in assuming that human γTuNA would also activate *Xenopus* γTuRC, as γTuNA is very well-conserved within the core region and only differing in the identities of four residues (87% identity; compare human aa sequence, “MKDFENQITELKKENFNLKLRIYFLEERMQQ” to *Xenopus* “MKDFE*K*QI*A*ELKKENFNLKLRIYFLEE*QV*QQ”). We chose to use the human sequence for our purification method as it appeared to bind *Xenopus* γTuRC with slightly greater affinity than the native sequence in our initial trials. In extract, both N-terminally Halo-tagged human and *Xenopus* γTuNAs block γTuRC activity to the same extent (see Figure 5 – Supplement 2A). This, coupled with previous work, suggests that the human and *Xenopus* γTuNAs appear to be largely interchangeable.

As we have directly demonstrated that γTuNA-bound γTuRC is an “activated” nucleator, this residual amount of γTuNA means that activity assays using a γTuNA-based purification method will have a greater baseline-level of γTuRC activity in the mock condition, as compared to those using γTuRC purified through other means. In effect, this means that our measurement of γTuNA’s effect on γTuRC activity (currently a 20-fold increase) is likely an undercount, with the true effect possibly being 30-fold or greater. The same would hold true for past measurements like those performed by Choi et al. and Muroyama et al. Most importantly, we do not believe this undercount, or elevated baseline, is an issue when it comes to demonstrating that dimers of γTuNA increase the number of microtubules nucleated by γTuRCs (i.e. activation). Rather this only slightly caps our measurement to less than might otherwise be possible, suggesting γTuNA might have an even more potent effect on γTuRC activity.

This important reviewer comment is now discussed in our revised manuscript on page 18 (lines 352-357).

2) In the title the authors use "Centrosomal motif, γ-TuNA" but I'm not sure it is fair to refer to it as a centrosomal motif, given that it is found in proteins that recruit γ-TuRCs to various MTOCs. CM1 refers to "Centrosomin motif 1", rather than centrosomal motif.

We thank the reviewer for their comment. We had been considering our findings in the context of the centrosome and as a base reconstitution of microtubule nucleation from the pericentriolar material where CDK5RAP2 is found in high concentration, hence our use of the centrosomal context in the title. However, we agree that our findings are broadly applicable to other MTOCs like the Golgi which also use γTuNA-containing proteins like CDK5RAP2 and myomegalin. We changed the title to use the “centrosomin” motif instead.

3) In the abstract, the authors say they "build on" their previous study, but I would argue that they are not building on it, they are correcting a major conclusion from it. Also, in the abstract they say that they "illuminate how γ-TuRC is controlled in space and time in order to build specific cytoskeletal structures" (and this statement is essentially repeated in the first line of the Discussion), but the authors have not studied these aspects at all. This, therefore, needs to be removed/toned down.

These are excellent points raised by the reviewer. We agree that in essence we are correcting a conclusion from our previous study concerning the role of γTuNA on γTuRC activity and consequently, we edited the abstract.

As to the spatial-temporal regulation of the level of γTuRC activity, we also agree that most of our data concerning the interaction of γTuNA with γTuRC, the effect on γTuRC activity, and the antagonistic role of stathmin are largely addressing the spatial regulation of microtubule nucleation (MTOC vs cytoplasm). However, previous work has shown that the number of CDK5RAP2 molecules at the centrosome (the most potent MTOC) increases during the transition from interphase to mitosis which results in increased MT nucleation activity via increased γTuRC recruitment (see Piehl et al., 2004; Lawo et al., 2012 and Mennella et al., 2012; and ). Thus, the temporal control of the number of γTuRC binding sites (in interphase vs mitosis) is a method of temporally regulating the level of microtubule nucleation, while simultaneously spatially restricting it. We see our findings here as supportive of the hypothesis that controlling the cellular location and number of γTuNAs over time is a means of spatially and temporally regulating γTuRC microtubule nucleation. On top of that, and critically, we directly show that we can immediately activate γTuRC activity via the addition of dimerized γTuNA, thus showing that the interaction of γTuRC and γTuNA at the centrosome are essentially an intertwined form of spatial and temporal control. To address these points, we edited the discussion in a section on spatio-temporal control (page 18).

4) It would be nice for the authors to improve the diagrams in Figure 1a/b, making it clearer where the sequence of γ-TuNA is located within the full length protein. They should include amino acid numbers in Figure 1b and also for all their constructs in Figure 1 supp 1 (not just 34aa fragment, etc). Is F75 really F75 in *Xenopus*? Which *Xenopus* isoform is being used?

We thank the reviewer for the suggested edits. We have modified Figure 1A to more clearly show where γTuNA is located in *Xenopus laevis* CDK5RAP2 (residues 56-86; isoform X1, 905 aa). This core γTuNA region is well-conserved when compared to the original, human γTuNA sequence first discovered by Fong et al., 2008 (human aa 58-90; isoform A, 1893 aa). Human γTuNA residues M60-Q90 are 87% identical to *Xenopus* residues M56-Q86. We have also changed Figure 1B to clearly show the alignment and conservation between the human and *Xenopus* γTuNA sequences, including adding their respective residue numbers. In this manner, we also highlight the well-conserved phenylalanine at position 75 in humans, which is well-conserved as phenylalanine 71 in *Xenopus.*

In using the human numbering scheme for these well-conserved residues, we hope to avoid confusing the reader when switching between species. Furthermore, as γTuNAs are present in both CDK5RAP2 and myomegalin within *Xenopus* and humans, and are present in homologs from other species, we believe using the well-conserved residue position in human CDK5RAP2 as a reference can help maintain comparability between past and future work. We alert the reader to this in lines 89 and 90, the legend to Figure 1, and lines 133-134.

5) Regarding F75A mutants: From the images in Figure 2B, and without seeing any quantification, it seems that there is some aster formation above the mock control when using the F75A fragments, suggesting that either the F75A mutant is capable of binding a low level of γ-TuRCs or that the F75A fragment binds something else that may promote a low level of MT nucleation. It seems unlikely that it F75A fragments are binding a low level of γ-TuRCs, given the data in Figure 3d/e. Based on the mass spec data, could γ-TuNA be allowing NME7 to bind? This could also explain why microtubule nucleation is higher with F75A than with other mutants in Figure 4B. I'm not expecting the authors to perform extra experiments for this, but a discussion would be nice.

We thank the reviewer for the comment. We agree that beads coated in F75A γTuNA mutant can weakly nucleate asters/MTs after incubation with extract, albeit at a qualitatively lower level than the wildtype γTuNA beads. We do note that in the Figure 2B image used for F75A γTuNA beads, there are other beads in the same field of view that are not nucleating asters as strongly as the central example (Figure 2B, F75A, upper right corner). We believe this does indicate that F75A does retain some weak, lowlevel ability to bind γTuRC. However, this ability does not seem strong enough to withstand stringent washing of these beads, which is where the discrepancy the reviewer noticed arises.

More specifically, in the Figure 2B experiment Ni-NTA beads were washed only once with 150 µL of buffer. For Figure 3D/E, we instead washed Halo beads two times with a total volume of 2 mL buffer, which reduced γTuRC signal to at or below the level of mock-treated beads. Furthermore, in the mass spectrometry experiment in Figure 2—figure supplement 1C, we had further increased the volume and stringency of these washes to match our γTuRC purification protocol, to the point that we do not detect γTuRC at all in either mock or F75A beads (Western blot, Figure 2—figure supplement 1C). Therefore, we conclude that the reason we can observe some small amount of γTuRC binding by F75A beads in Figure 2B (and slight increase in extract activity in Figure 4B), but not detect this in our Westerns (Figure 3D/E, Figure 2—figure supplement 1C) is solely due to the presence of these different stringency wash steps.

Furthermore, we hypothesize that as mutating F75 residue does not impair γTuNA dimerization (Figure 3B), most of the γTuRC binding interface is preserved and thus can still possibly transiently interact with γTuRC. However, without the two flanking phenylalanine residues at position 75 (Figure 3A), this dimer cannot remain firmly bound to γTuRC and dissociates. This would explain why, at high concentration on beads, F75A can retain a diminished amount of associated γTuRCs, but not strongly enough to withstand subsequent washing.

Finally, we have no evidence to suggest that γTuNA might allow or help NME7 to bind, but we are excited by the possibility that a γTuNA-bound γTuRC might form a novel interface that can help subsequent binding of other factors. However in this specific case, prior work has firmly demonstrated that NME7 is a γTuRC subunit that is present regardless of how γTuRC is purified or whether γTuNA is present (Hutchins et al., *Science*, 2010; Teixido-Traversa et al., *Mol Biol Cell*, 2010; Liu et al., *Mol Biol Cell*, 2014; Liu et al., *Nat,* 2020; Consolati et al., *Dev Cell*, 2020; Wieczorek et al., *Cell*, 2020, and this work). Thus NME7’s presence in γTuRC is likely independent of γTuNA.

6) The authors discuss the idea that the effect of γ-TuNA could be further enhanced by γ-TuRCs being recruited to sites of high local tubulin concentration. Again, this would be easy for them to test, by varying tubulin concentration in their assays and measuring the fold change in γ-TuRC-mediated nucleation in mock vs γ-TuNA addition.

We thank the reviewer for their comment**.** We agree that varying the tubulin concentration is a worthwhile experiment, especially in light of our hypothesis that γTuNA enables γTuRC to nucleate MTs at lower tubulin concentrations than otherwise possible (as suggested by our in vitro experiments using stathmin in Figure 6).

While we could increase the tubulin concentration above 15 µM tubulin in the in vitro assay, we do not report higher concentrations in this study as background spontaneous MT nucleation increases such that it can become difficult to distinguish γTuRC-nucleated MTs. Also, at 15 µM tubulin and 1.65 µM γTuNA dimer (3.3 µM monomer γTuNA) we find our assay saturates within 25 seconds, by which time we see a maximum of ~300 MTs per 40 µm by 40 µm area on the coverslip (Figure 5C). This is in agreement with the number of fluorescent γTuRCs we see for the equivalent surface area (mean: 377 MTs; first to third quartile range: 302 to 469 γTuRCs; see Author Response image 1). For this reason, we believe that practically all nucleation-competent γTuRCs attached to our coverslips are nucleating MTs within 25 seconds in the presence of 15 μm tubulin and 1.65 µM γTuNA dimer (3.3 µM monomer). Thus 15 µM is the highest we can increase the tubulin concentration in our assays.

As a result, we decided to instead decrease our tubulin concentration in the assay, setting it at γTuRC’s critical concentration. We then assayed for γTuNA’s effect on γTuRC MT nucleation at this low tubulin concentration in vitro (now incorporated into Figure 6, as panel E). In our previous study (Thawani et al., 2020), we found that the critical concentration in the presence of γTuRC was 7 µM tubulin, where γTuRC has very low but still detectable nucleation ability (at least 1 MT nucleated per field during the course of a 5 min experiment). This was also observed by our colleagues in Consolati et al., *Dev Cell*, 2020 at 7.5 µM tubulin (defined by the authors as at least 1 MT nucleated per 164 µm^2^ field over 20 min). As shown in Figure 6E, we again validate this finding, observing that our purified γTuRC has very little activity at 7 µM tubulin with only 1 to 2 MTs nucleated per field of view over the course of 5 min (mock buffer condition). These MTs are also short and appear to be more dynamic than those generated by γTuRC at 15 µM tubulin. In Figure 6E, we show all MTs nucleated in our assay at 7 µM tubulin, regardless of lifetime, by presenting max intensity projections for our 5 min time series. Critically, in the presence of 1.65 µM γTuNA dimer (3.3 µM monomer), ~40 γTuRCs nucleate MTs per field despite being at this low tubulin concentration. This is approximately a 25-fold increase in the total number of MTs generated over the entire 5 min assay, as compared to the buffer condition (Figure 6E). Thus, we find further validation that γTuNA enables γTuRC to nucleate MTs even under low, constrained tubulin concentrations, an observation we first described in our experiments with stathmin (Figure 6A-D).

We would also like to note that this new experiment is conceptually equivalent to our Figure 6C experiments using 15 µM tubulin and 4 µM stathmin. As 1 molecule of stathmin sequesters 2 tubulin subunits, 4 µM stathmin would leave only ~7 µM tubulin free in a 15 µM tubulin reaction (assuming 100% of the stathmin is functional). The fact that these two experiments agree not only confirms that γTuNA increases γTuRC’s efficiency (i.e. lowers γTuRC’s critical concentration), but also that stathmin indirectly regulates γTuRC activity by constraining the cytoplasmic tubulin pool.

**Author response image 1. sa2fig1:** Analysis of the number of γTuRCs present in the in vitro TIRF assays. (A) Image analysis of fluorescent γTuRC (labeled with NHS-Alexa 568) in an in vitro TIRF assay. Unlike in assays shown in Figure 5, γTuRC was further diluted two-fold with buffer to aid in accurate counting. Images were binarized and counted via ImageJ’s “Analyze Particles” function. Outlines of counted complexes are shown in “A” rightmost panel. Arrows mark complexes that the automatic analysis undercounted as a single γTuRC. Without dilution of γTuRC, this undercounting is worsened and leads to significant error. (B) Quantification of the number of γTuRC spots, corrected for dilution factor. The mean, marked by an “X”, is 377 γTuRCs (±64 SEM). The first to third quartile range (red box) is 302 to 469 γTuRCs.

Reviewer #2 (Recommendations for the authors):Overall, I think this is a well-controlled study. However, the discussion that emphasizes the importance of γ-TuNA-dimerization, as a pre-requisite for the binding and activation of γ-TuRCs, has some flaws that should be addressed:1) The human double mutant I67A/L70A has a strong negative effect on γ-TuNA dimerization (Figure 3B), but still enables the binding to γ-TuRCs (Figure 3E).2) The mutant F75A is perfectly able to dimerize, but fails to bind γ-TuRCs and fails to stimulate microtubule nucleation.These discrepancies are neglected in the results and Discussion sections. They suggest that failure of dimerization may be overcome by certain mutations such as I67A/L70A, and they suggest that the γ-TuNA external protein surface (not involved in the coiled-coil) is also essential for the binding to the γ-TuRC.

We thank the reviewer for their comment**.** In regard to the first point, we agree that the human I67A/L70A mutant has a strong negative effect on dimerization, but still retains some γTuRC binding just below the level of the L70A single point mutant (Figure 3E). If this is compared to human I67D/L70D (Figure 3F) we can see that double substitution to aspartate, instead of alanine, completely removes this residual γTuRC binding. This suggests that retaining some hydrophobicity at these positions might preserve enough of the coil structure to allow for a weak interaction with γTuRC, despite lacking the required hydrophobicity to first form the coiled-coil dimer (Figure 3A).

Given that, how might this I67A/L70A mutation be overcoming the loss of dimerization to weakly bind γTuRC? It is possible that two separate monomeric coils of mutant γTuNA might bind the same γTuRC and form a stable complex. In this scenario, the interaction with the γTuRC would stabilize the γTuNA dimer, overcoming the loss of the strongly hydrophobic contacts normally present in the coiled-coil dimer interface. We believe that we have observed this phenomenon with the *Xenopus* L77A mutant in our in vitro reactions (Figure 5B), where late in the assay L77A can begin to increase γTuRC activity despite lacking complete dimerization and strong γTuRC binding ability (Figure 3B-E). We predict that this can only occur in hydrophobic-to-weaker-hydrophobic substituted versions of γTuNA, like I67A/L70A or L77A. These types of substitutions are not as likely to cause drastic changes to the overall coil structure of a γTuNA monomer, which might allow for two of these monomers to be stabilized into a dimer on γTuRC.

We believe it is for this reason that the aspartate substituted version, I67D/L70D, cannot bind γTuRC as a switch from hydrophobicity to hydrophilicity is likely too drastic a change for the coil structure. In support of this, closer inspection of the peak SEC retention volumes in Figure 3B-C reveals that human I67D/L70D is eluted ahead of human I67A/L70A (~15.0 vs ~15.2 mL), indicating that I67A/L70A has a smaller hydrodynamic radius despite differing in only two residues. We believe this difference is reflective of changes in the γTuNA coil structure, where the hydrophilic aspartate residues now cause the coil to extend, kink, or otherwise deform in a way that increases the hydrodynamic radius of the I67D/L70D protein. This change in structure, in addition to blocking coiled-coil dimerization, also likely prevents even weak interactions with γTuRC thereby blocking the γTuRC-induced dimer scenario outlined above.

As to the second point, we completely agree that our data concerning the F75A mutant suggests the external surface of the γTuNA coiled-coil dimer is critical for stable γTuRC interaction. That was our interpretation as well, with both dimerization and the F75 residue being critical for γTuRC interaction. We stated this conclusion in the title for Figure 3 and the section title on line 142. Moreover, we have added an additional line to the Results section and another line in the discussion to stress this point. We believe that the F75 residue is critical for “locking” the dimer into γTuRC, and have discussed this in our response to Reviewer #1 (point #5) as well.

Reviewer #3 (Recommendations for the authors):The present study focuses on the effect of a specific structural motif γ-TuNA in microtubule formation. The study is an important advance to widen the discussion around the spatiotemporal regulation of microtubule nucleation through γ-TuNA-motif-containing proteins. Although the activating function of γ-TuNA was reported earlier, there were contrasting results in the literature that have been resolved by this study. Another advance is a scaled-up method to purify γTuRC using Halo- γ-TuNA and validation of the purification through in vitro nucleation experiments and electron microscopy. While the study does not provide new mechanistic insights to understand how γ-TuNA affects γTuRC and whether the observed effects on nucleation are a general feature of γ-TuNA motifs in the context of different proteins such as CDK5RAP2, TPX2, and CEP215, the work here is important for the field as suitable for publication as an eLife advances report. The following needs to be addressed in the text prior to publication.1. The authors make a strong claim that the dimeric state of γ-TuNA is necessary for binding γ-TuRC and activating microtubule nucleation. Although the authors show that the double and triple mutants show loss of dimerization and binding to γ-TuRC, there are multiple exceptions that cannot be explained and should be discussed explicitly:a. L70A mutant does not fully dimerize (comparable to L77A) but can still bind γ-TuRC (Figure 3).b. GCN4-tagged mutants that form a dimer cannot bind γ-TuRC (Fig-3-Suppl-1).c. The most important of all exceptions is the L77A mutant that is neither a complete dimer, nor does it bind γ-TuRC (Figure 3) but is still able to activate microtubule nucleation at longer time points (Figure 5).d. Related to (c), the authors state that "Interestingly, we found that the intermediate dimer mutants (I67A, L70A, or L77A) had correspondingly intermediate levels of γTuRC" (line 149), but the dimerization capability of L70 and L77 is similar and yet very different in γ-TuRC binding.e. L70A is missing in Figure-5 and having that information will be helpful in making correlative conclusions (NOT ESSENTIAL).

We thank the reviewer for their comment. Below are our responses to each point in #1.

Points 1a and 1d We agree that the L70A and L77A mutants have similar impairments in dimerization (Figure 3B), but divergent γTuRC binding ability in extract pulldowns (Figure 3E). We believe this suggests that the region of the γTuNA coil from position 75 to position 77 (F75, L77) is the core γTuNAγTuRC binding interface, within which mutations are not well-tolerated for stable γTuRC interaction in extract. We note that mutations at positions moving from this core towards the N-terminus (L70 to I67 to F63) have less and less impact on both dimerization and γTuRC binding ability in extract. Since the position within the coil appears to matter for γTuRC interaction, this could explain the divergent behavior for L70A and L77A, as L70 appears to be outside the most critical region and thus can still retain a small amount of γTuRC binding.

Points 1c and 1e We agree that at a late stage L77A can increase MT nucleation by γTuRC in vitro (Figure 5). In our response to Reviewer #2, we discuss how impaired dimerization might be rescued by two γTuNA monomers interacting with γTuRC separately. This interaction with γTuRC could theoretically form a stable complex (with a pseudo-dimer of L77A γTuNA) given enough time and high concentration of monomers. We believe this explains why L77A’s effect is greatly delayed (onset >150 sec; Figure 5B) compared to wildtype (onset < 10 sec). However, as we do not observe L77A binding γTuRC or triggering activation in extract, even at five minutes, we conclude that non-specific competition from extract factors prevents this late-stage activation from occurring. That is to say, the conditions in vitro are permissive of interactions that might otherwise not be strong or efficient enough to overcome barriers present in cytoplasm. The fact that wildtype γTuNA does so regardless of whether it is in extract or in vitro demonstrates the strength of the affinity for γTuRC.

As to the design of our in vitro assay, our main goal was to reconstitute γTuRC activation by wildtype γTuNA and observe this directly in real-time. However, we included the F75A and L77A mutants to also dissect the impact on γTuRC activation caused by affecting the dimer versus affecting the external dimer surface. We chose L77A to represent impaired dimerization (over F63A, I67A, or L70A) as it is the most disruptive single point mutation that affects both dimerization and γTuRC binding in extract. F75A is the most disruptive mutation for γTuRC binding that has no effect on dimerization. Comparing these to wildtype allowed us to conclude that both dimerization and the F75 residue are required for immediate activation of γTuRC. We predict that F63A, I67A, and L70A will also activate γTuRC in vitro, but we did not perform those experiments due to the significant effort involved in these in vitro assays.

Point 1b The γTuNA-GCN4 fusion constructs were our attempt to rescue the coiled-coil dimer and subsequent γTuRC binding ability (Fig-3-Suppl-1). While this did rescue dimerization in the SEC assay, we concede that we don’t know if GCN4-induced dimerization properly restores the natural coiledcoil γTuNA dimer structure. If not (as seems to be the case), this artificial dimer would not be bound by γTuRC. Also, we would suggest that specific residues might be required for both dimerization and for making specific contacts with γTuRC. For these cases, inducing dimerization would never be sufficient to restore wildtype levels of γTuRC binding as the specific residue enabling stable interaction would still be missing. We stress that dimerization is a key component of how γTuNA interacts with γTuRC (and supported by the cryo-EM structure by Wieczorek et al., *Cell Reports*, 2020), but dimerization is not on its own sufficient. Having specific residues in the coil, like F75, is also required (Figure 3 and 5). We imagine that some residues, like L77, have impacts on both dimerization and stable γTuRC binding.

Overall, I am convinced that the residues in the coiled-coil are important for γ-TuRC binding but not fully convinced about the dimerization claim without an experiment where forced dimerization restores activity in a mutant. The writing should be edited for greater precision.2. The study discusses that the effect of γ-TuNA on γTuRC is transient and might depend on local tubulin concentrations. Related to this point, the authors explicitly show in their model that γ-TuNA counteracts the effects of stathmin by increasing the efficiency of nucleation at lower tubulin concentrations (Fig6). This hypothesis can be easily tested in their in vitro assay by examining if γ-TuNA decreases the critical conc. of tubulin required for γ-TuRC-based microtubule nucleation. This would provide a mechanism for γ-TuNA's effects and strengthen the paper. If this cannot be done, then I suggest removing this specific mechanistic detail from the model figure.

We thank the reviewer for their comment and suggested experiment. We agree that testing whether γTuNA increases γTuRC’s efficiency at lower tubulin concentrations is worthwhile, especially in light of our observation that stathmin’s indirect repression of γTuRC MT nucleation can be overcome by addition of γTuNA (Figure 6). We performed this experiment and present it in Figure 6, as panel E. As we discuss in greater detail in our response to Reviewer #1 (point #6), we repeated our in vitro assays at 7 µM tubulin, which we and others have found to be γTuRC’s critical concentration (Thawani et al., *eLife*, 2020; Consolati et al., *Dev Cell*, 2020). At this low concentration, γTuRCs rarely nucleate MTs (~1.5 MTs over the entire 5 min assay). However, in the presence of 1.65 µM wildtype γTuNA dimer (or 3.3 µM monomer), ~40 γTuRCs nucleate (25-fold increase in activity). Thus, γTuNA increases γTuRC’s efficiency at lower tubulin concentrations. An equivalent statement would be that γTuNA decreases γTuRC’s criticial tubulin concentration.

3. The γ-TuRC purified using γ-TuNA-coated beads followed by sucrose gradient step still retains ~30% wild type γ-TuNA (methods Line 486-488). This should be explicitly stated in the main text as it is very relevant to experiments where mutant γ-TuNA is added, and the effects observed are on top of the wild-type γ-TuNA from the purification.

We thank the reviewer for the comment. We have added an explicit statement about the background γTuNA level and how this increases the baseline activity of γTuRC in the assays (see lines 352-357; also discussed in our response to Reviewer #1, point #1.)

4. In Figure 1F, the authors deplete γTuRC from egg extracts and conduct microtubule nucleation experiments to show the dependence of γ-TuNA on the γTuRC based templates to build microtubules. Although this experiment was effective in proving the point, the authors should explain their decision to choose γ-TuNA coupled beads to deplete γTuRC from the extracts. Depleting γTuRC using any other mechanism, for example, anti-γ tubulin antibody coupled beads should also show similar effects.

We thank the reviewer for the comment. We initially did perform both our γTuRC depletions and purifications with a homemade rabbit γ-tubulin antibody (same antigen as was used in Thawani et al., *eLife*, 2020). However, our antibody suffered from batch-to-batch variability, which meant reproducing a near-complete depletion of γTuRC was difficult. As we had great success using Halo-γTuNA to purify γTuRC from extract with consistent depletion, we decided this would be the most reproducible method for testing whenter γTuRC was required for γTuNA’s activation effect in extract. We have no doubt that immunodepleting γTuRC with a well-behaved antibody would also demonstrate this.

5. The authors write in the discussion that γ-TuNA fails to activate antibody purified γTuRC. This is extremely puzzling. Does a comparison of the two preps γTuRC (antibody-based purification and γTuNA-based purification) by mass spec provide any hints to additional factors? If there was an additional activator in the γ-TuNA-based purification, then the effects observed in this paper are not entirely due to the γ-TuNA.

We thank the reviewer for the comment. We agree that this inability to activate antibody purified γTuRC was puzzling for us. We initially thought, as the reviewer suggests, that an additional factor might be required for this γTuNA-based activation. We originally set out to find this additional factor. However, even with mass spectrometry data from our group and others (Liu et al., 2020; Consolati et al., 2020; Wieczorek et al., 2020), we do not find an obvious target. Furthermore, silver stain of the peak γTuRC fraction for our prep showed the same banding pattern as that published with our previous antibody prepped γTuRC, indicating that aside from γTuRC components, there was no other major contaminant or factor present to explain the response to γTuNA. Rather, we believe the difference can be explained by the greater yield and consistent quality of γTuRC provided by the Halo-γTuNA prep.

6. The authors utilize purified γ-TuNA motif from CDK5RAP2 in this study. The authors indicate that γ-TuNA motif has no role in γTuRC assembly through fractionation (pg, 4, line 90-92 and Figure 1-Suppl-2), drawing a clear distinction between γ-TuNA and other nucleation activators. However, in the context of full-length CDK5RAP2, the γ-TuNA motif may very well have a role in γTuRC assembly. So, this conclusion seems like a stretch. I recommend changing the text. Related to this, ideally "γ-TuNA motif or domain" would be a better phrase choice throughout the paper instead of γ-TuNA.

We thank the reviewer for the comment. For CDK5RAP2 in humans and frogs, we do not know of any prior work or evidence that might suggest its involvement in γTuRC assembly. The closest CM1-containing proteins for which this seems to be true are the yeast proteins Spc110p and Spc72p (Kollman et al., *NSMB*, 2015; Lyon et al., *MBoC*, 2016), where γ-TuSC subunits are oligomerized by several CM1 motifs into γTuRCs (Brilot et al., 2021). Rather, recent work has shown that in vertebrates γTuRC is assembled via the RUVBL1-RUVBL2 AAA ATPase complex (Zimmermann et al., *Sci. Adv.,* 2020), independently of any CM1-containing protein. In fact, heterologous expression of the core γTuRC subunits and the RUVBL1/2 complex in insect cells is sufficient to assemble intact human γTuRCs. We believe this supports our observation that the presence of high concentrations of γTuNA in *Xenopus* extract ultimately have no effect on γTuRC assembly. Of course one cannot exlude at this point that full-length CDK5RAP2 could provide additional help for γTuRC ssembly, and that would remain to be tested.

7. In Fig-2-supp-1-B, please label the bands on the gel that are of interest.

We thank the reviewer for the comment. We have labeled the gel with bands of interest.

[Editors’ note: further revisions were suggested prior to acceptance, as described below.]

Reviewer #1 (Recommendations for the authors):New Reviewer Response:I appreciate the authors additional section in the discussion about the work that others have carried out on the spatiotemporal control of γ-TuRC activation. However, it seems the authors have misunderstood my original point about not showing data on the spatiotemporal regulation of γ-TuRC activation – I did not mean that they had only addressed spatial control; I meant that they had not addressed either spatial or temporal control. This is because their data shows, very nicely, that γ-TuNA binding to γ-TuRCs activates γ-TuRCs but it does not show how γ-TuNA binding to γ-TuRCs is regulated in space and time. The authors appear to propose that spatial control is achieved by localising CDK5RAP2 to centrosomes, but CDK5RAP2 is also present in the cytoplasm and so, in theory, has the potential to bind (and activate) γ-TuRCs in the cytoplasm. As the authors discuss, this binding is likely auto-inhibited by other cis-regulatory elements, but this idea comes from other studies, not this one. And they certainly have not linked this to building "specific cytoskeletal structures". I think the authors should focus their summary statement on what they have actually shown – something more like "In sum, our improved assays finally prove that γ-TuNA binding strongly activates γ-TuRCs, explaining previously observed effects of γ-TuNA expression in cells".

Thank you for the clarification. We have removed all discussion of spatio-temporal regulation, except for a single mention of the CM1/γTuNA phosphorylation studies in lines 427428. Within that context, we have also added a mention that we cannot rule out cytoplasmic activation (as noted by the reviewer), lines 424-428.

We have also edited the abstract to use the reviewer’s suggested edit (lines 18-20).

New Reviewer Response:I appreciate the extra experiments, but I'm not sure they address the initial point, which was whether "the effect of γ-TuNA could be further enhanced by γ-TuRCs being recruited to sites of high local tubulin concentration", and not whether γ-TuNA promotes microtubule nucleation at low tubulin concentrations (which it clearly does). The authors actually make the point that raising the tubulin concentration in their experiments makes it hard to distinguish the microtubules nucleated by γ-TuRCs from those nucleated spontaneously. From this, I would assume that the effect of γ-TuNA would not be further enhanced when γ-TuRCs are recruited to sites of high local tubulin concentration, because microtubules would simply spontaneously nucleate if the tubulin concentration were high enough. It seems more valid to report that γ-TuNA is required when tubulin concentrations are low.

Thank you for the clarification. As the reviewer has suggested, we have removed our prediction that gTuNA’s effect on γTuRC might be enhanced at high local tubulin concentrations. Instead, we report that γTuNA stimulates γTuRC even when tubulin concentrations are low (lines 364-365).

Reviewer #2 (Recommendations for the authors):As I stated already in my earlier review, I am in favor of this manuscript.I am a bit disappointed, though, by the manner that this revision has been performed: the authors have provided detailed responses to my concerns in their rebuttal letter (importance of dimerization for gTuRC-binding; gTuNA mutants), but it seems that they haven't significantly addressed these points in the revised manuscript. Similarly, they ignored to discuss related concerns raised by reviewer 3. Our comments were meant to help improving the manuscript! Simply arguing in the rebuttal is not enough…

Thank you for the clarification and for the prompt to address these points in the manuscript even further. We have replaced the Discussion sub-section on spatio-temporal regulation in favor of a new section explicitly discussing the divergent mutant behaviors and the importance of dimerization to γTuRC binding (lines 447-501). In reviewer 2’s previous comments, they specifically mentioned mutant I67A/L70A and mutant F75A. I67A/L70A is discussed in lines 458-486, while F75A is discussed in lines 497-501. We apologize for not explicitly incorporating them in the last revised manuscript.